# Silicone oil-induced ocular hypertension and glaucomatous neurodegeneration in mouse

Jie Zhang[1,2†], Liang Li[1†], Haoliang Huang[1], Fang Fang[1,3], Hannah C Webber[1], Pei Zhuang[1], Liang Liu[1], Roopa Dalal[1], Peter H Tang[1,4], Vinit B Mahajan[1,4], Yang Sun[1,4], Shaohua Li[2], Mingchang Zhang[2], Jeffrey L Goldberg[1], Yang Hu[1]*

[1]Department of Ophthalmology, Stanford University School of Medicine, Palo Alto, United States; [2]Department of Ophthalmology, Tongji Medical College, Union Hospital, Huazhong University of Science & Technology, Wuhan, China; [3]Department of Ophthalmology, The Second Xiangya Hospital, Central South University, Changsha, China; [4]Department of Ophthalmology, Veterans Affairs Palo Alto Health Care, Palo Alto, United States

**Abstract** Understanding the molecular mechanism of glaucoma and development of neuroprotectants is significantly hindered by the lack of a reliable animal model that accurately recapitulates human glaucoma. Here, we sought to develop a mouse model for the secondary glaucoma that is often observed in humans after silicone oil (SO) blocks the pupil or migrates into the anterior chamber following vitreoretinal surgery. We observed significant intraocular pressure (IOP) elevation after intracameral injection of SO, and that SO removal allows IOP to return quickly to normal. This simple, inducible and reversible mouse ocular hypertension model shows dynamic changes of visual function that correlate with progressive retinal ganglion cell (RGC) loss and axon degeneration. It may be applicable with only minor modifications to a range of animal species in which it will generate stable, robust IOP elevation and significant neurodegeneration that will facilitate selection of neuroprotectants and investigating the pathogenesis of ocular hypertension-induced glaucoma.
DOI: https://doi.org/10.7554/eLife.45881.001

*For correspondence:
huyang@stanford.edu

†These authors contributed equally to this work

**Competing interests:** The authors declare that no competing interests exist.

## Introduction

Glaucoma is the most common cause of irreversible blindness and will affect more than 100 million individuals between 40 and 80 years of age by 2040 (*Tham et al., 2014*). Annual direct medical costs to treat this disease in 2 million patients in the United States totaled $2.9 billion (*Varma et al., 2011*). Glaucoma is a neurodegenerative disease characterized by injury to the axons of retinal ganglion cells (RGCs) followed by progressive degeneration of RGC somata and axons within the retina and Wallerian degeneration of the myelinated axons in the optic nerve (ON) (*Quigley, 1993*; *Quigley et al., 1995*; *Libby et al., 2005*; *Howell et al., 2007*; *Weinreb and Khaw, 2004*; *Calkins, 2012*; *Burgoyne, 2011*; *Nickells et al., 2012*; *Jonas et al., 2017*). The level of intraocular pressure (IOP) is the most common risk factor (*Singh and Shrivastava, 2009*). Current clinical therapies target reduction of IOP to retard glaucomatous neurodegeneration (*The AGIS Investigators, 2000*; *Early Manifest Glaucoma Trial Group et al., 2002*; *Lichter et al., 2001*), but neuroprotectants are critically needed to prevent degeneration of RGCs and ON. Similar to other chronic neurodegenerative diseases (*Varma et al., 2008*), the search for neuroprotectants to treat glaucoma continues. To longitudinally assess the molecular mechanisms of glaucomatous degeneration and the efficacy of

neuroprotectants, a reliable, reproducible, and inducible experimental ocular hypertension/glaucoma model is essential.

The rodent serves as the mammalian experimental species of choice for modeling human diseases and large-scale genetic manipulations. Various rodent ocular hypertension models have been developed including spontaneous mutant or transgenic mice and rats and mice with inducible blockage of aqueous humor outflow from the trabecular meshwork (TM) (*Pang and Clark, 2007*; *Morrison et al., 2008*; *McKinnon et al., 2009*; *Chen and Zhang, 2015*). While genetic mouse models are valuable to understand the roles of a specific gene in IOP elevation and/or glaucomatous neurodegeneration, the pathologic effects may take months to years to manifest. Inducible ocular hypertension that develops more quickly and is more severe term would be preferable for experimental manipulation and general mechanism studies, especially for neuroprotectant screening. Injection of hypertonic saline and laser photocoagulation of the episcleral veins and TM are commonly used in rats and larger animals (*Morrison et al., 2008*). Although similar techniques also produce ocular hypertension in mice (*Aihara et al., 2003*; *Grozdanic et al., 2003*; *Yun et al., 2014*), they are technically challenging, and irreversible ocular tissue damage and intraocular inflammation complicate their interpretation (*Pang and Clark, 2007*; *Chen and Zhang, 2015*). Intracameral injection of microbeads to occlude aqueous humor circulation through TM produces excellent IOP elevation and glaucomatous neurodegeneration (*Sappington et al., 2010*; *Chen et al., 2011*; *Cone et al., 2010*; *Samsel et al., 2011*). However, retaining microbeads at the angle of the anterior chamber and controlling the degree of aqueous outflow blockade are difficult. Furthermore, its lengthy duration (6–12 weeks after microbeads injection) causes death of only less than 30% of RGC (*Cone et al., 2010*; *Ito et al., 2016*; *Yang et al., 2016*), leaving a narrow window for preclinical testing of neuroprotective therapies. It is therefore critically important to develop a simple but effective ocular hypertension model in mice that closely resembles human glaucoma, and that can be readily adapted to larger animals with minimal confounding factors.

Secondary glaucoma with acutely elevated IOP occurs as a post-operative complication following the intravitreal use of silicone oil (SO) in human vitreoretinal surgery (*Ichhpujani et al., 2009*; *Kornmann and Gedde, 2016*). SO is used as a tamponade in retinal detachment repair because of its buoyancy and high surface tension. However, SO is lighter than the aqueous and vitreous fluids and an excess can physically occlude the pupil, which prevents aqueous flow into the anterior chamber. This obstruction increases aqueous pressure in the posterior chamber and displace the iris anteriorly, which causes angle-closure, blockage of aqueous outflow through TM, and a further increase in IOP. Prophylactic peripheral iridotomy that maintains the circulation between anterior and posterior chambers normally prevents this type of secondary glaucoma. Based on this clinical experience, we developed a simple procedure for intracameral injection of SO to block the pupil, which causes acute ocular hypertension and significant RGC and ON degeneration. The present study demonstrates that this model, which may be adaptable to larger species, induces stable IOP elevation and profound neuronal response to ocular hypertension in the retina that will expedite selection of neuroprotectants and establishing the pathogenesis of acute ocular hypertension-induced glaucoma.

## Results

### Intracameral SO injection induces ocular hypertension by blocking the pupil and aqueous humor drainage

Although intravitreal injection of SO in vitreoretinal surgeries can cause post-operative secondary glaucoma in humans (*Ichhpujani et al., 2009*; *Kornmann and Gedde, 2016*), we reasoned that direct injection of SO into the anterior chamber of mice would be more efficient, preventing the need to remove the vitreous and reducing toxicity due to direct contact with the retina. As illustrated in *Figure 1A,B* and *Video 1*, after intracameral injection SO forms a droplet in the anterior chamber that contacts the surface of the iris and tightly seals the pupil due to high surface tension. To test whether SO blocks migration of liquid from the back of the eye to the anterior chamber, we injected dye (DiI) into the posterior chamber and visualized its migration into the anterior chamber. In dramatic contrast to a normal naïve eye, in which copious dye passed through the pupil and appeared in the anterior chamber almost immediately after injection, no injected dye reached the

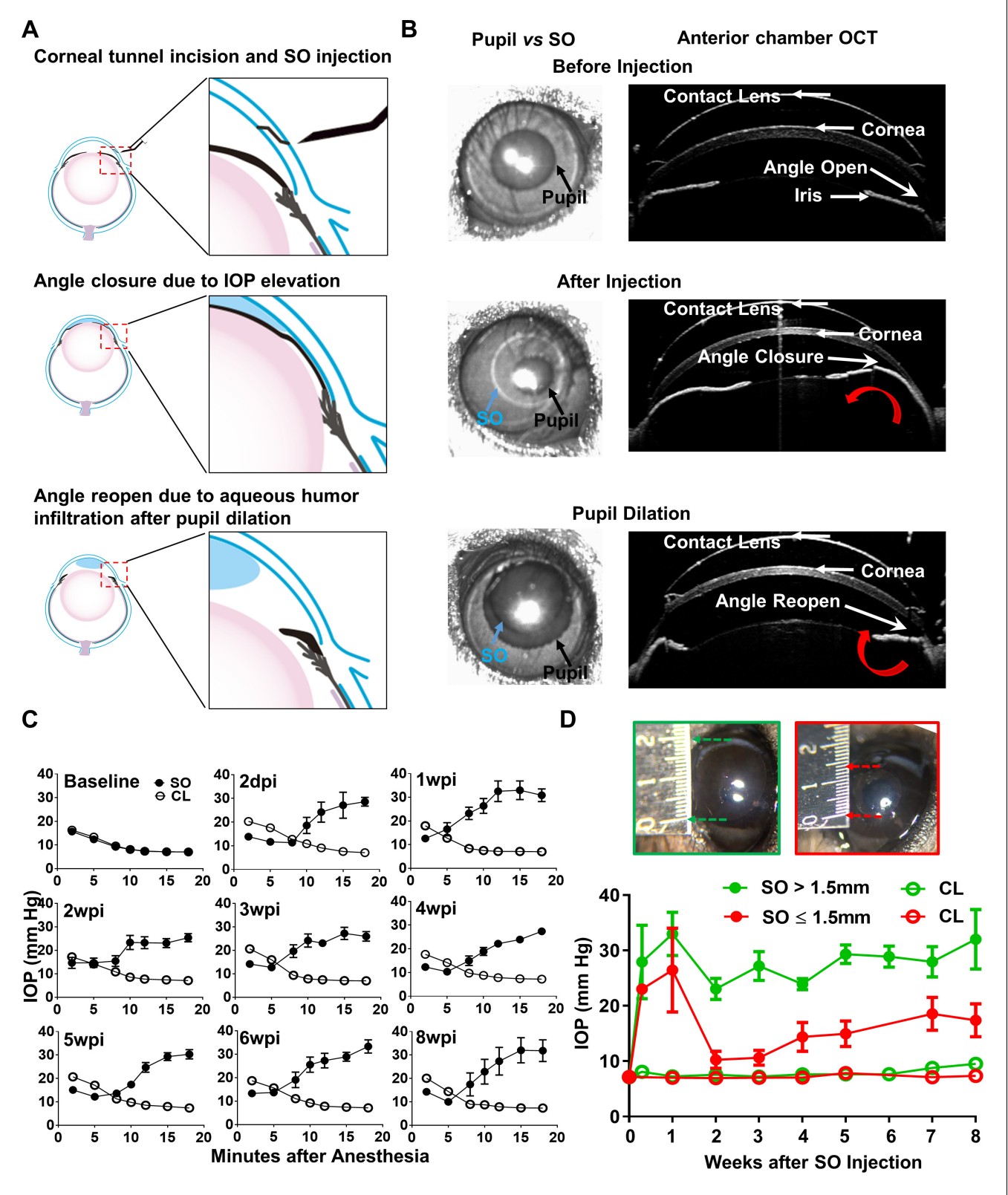

**Figure 1.** Silicone oil-induced ocular hypertension under-detected (SOHU) mouse model. (**A**) Cartoon illustration of SO intracameral injection, pupillary block, closure of the anterior chamber angle, and reopening of the angle of anterior chamber after pupil dilation. (**B**) Representative anterior chamber OCT images of SOHU eyes in living animals showing the relative size of SO droplet (blue arrow) to pupil (black arrow) and the corresponding closure or opening of the anterior chamber angle before and after pupil dilation. Red curved arrow indicates the direction of aqueous humor flow. (**C**)

*Figure 1 continued on next page*

*Figure 1 continued*

Longitudinal IOP measurements at different time points before and after SO injection, and continuous measurements for 18 min after anesthesia with isoflurane at each time point. (D) The sizes of SO droplet and corresponding IOP measurements at different time points after SO injection; IOP measured 12–15 min after anesthesia. SO: SO injected eyes; CL: contralateral control eyes. Data are presented as means ± s.e.m, SO > 1.5 mm, n = 17; SO ≤ 1.5 mm, n = 6.

DOI: https://doi.org/10.7554/eLife.45881.002

The following figure supplement is available for figure 1:

**Figure supplement 1.** Extended pupillary dilation lowers down IOP in the SOHU eyes.

DOI: https://doi.org/10.7554/eLife.45881.003

anterior chamber of the SO eye (*Videos 2* and *3*). This result indicates that SO causes effective pupillary block.

The ciliary body constantly produces aqueous humor, which accumulates in the posterior chamber and pushes the iris forward. When the iris root touches the posterior corneal surface, the anterior chamber angle closes (*Figure 1A*), as evidenced by live anterior chamber optical coherence tomography (OCT) (*Figure 1B*). The angle closure can further impede the outflow of aqueous humor through TM and may also contributes to IOP elevation. Dilation of the pupil until it is larger than the SO droplet can relieve the pupillary block. *Video 4* shows that after pupil dilation aqueous humor floods into the anterior chamber and pushes the SO droplet away from the iris, which reopens the anterior chamber angle (*Figure 1A,B*). Together, these results characterize the series of reactions initiated by intracameral SO injection, including the physical mechanisms of SO-induced pupillary block, posterior accumulation of aqueous humor, peripheral angle-closure, and IOP elevation.

We measured the IOP of the experimental eyes once weekly for 8 weeks after a single SO injection and the contralateral control (CL) eyes after a single normal saline injection. Surprisingly, IOP was lower in the SO eyes than in CL eyes when measured immediately after anesthetizing the animals with isoflurane (*Figure 1C*). The TonoLab tonometer used to measure mouse IOP is based on a rebound measuring principle that uses a very light weight probe to make momentary contact with the center of the cornea, which primarily measures the pressure of anterior chamber. Measurements over extended periods of time showed the IOP of the SO eyes to be progressively and significantly elevated, in dramatic contrast to the CL eyes, in which IOP decreased over time. The increasing IOP in the SO eyes closely correlated with the change in pupillary size, indicating a significant role of pupillary block. Pupillary dilation removed the pupillary block and allowed the tonometer to detect higher IOP after aqueous humor migration into the anterior chamber, which reflects the elevated IOP in the posterior segment of the eye. Pupillary size reached its maximum and IOP reached to its plateau about 12–15 min after induction of anesthesia with continuous isoflurane inhalation. In mice in which we measured IOP for as long as 30 min under anesthesia, however, the IOP eventually declined, indicating effective TM clearance of aqueous during this time (*Figure 1—figure supplement 1*). Therefore, we standardized the time period (12–15 min after induction of anesthesia) for measuring IOP in later experiments. Because the unique feature of this novel experimental glaucoma model is that the ocular hypertension is under-detected in non-dilated eyes, we named it 'SO-induced ocular hypertension under-detected (SOHU)".

IOP elevation in the SO eye started as early as 2 days post injection (2dpi) and remained stable for at least 8 weeks (the longest time point we tested) at an IOP about 2.5 fold that of CL eyes, if the diameter of the SO droplet was larger than 1.5 mm (*Figure 1D*). We achieved this size of SO droplet in about 80% of mice, but in the 20% of mice with a small SO droplet (≤1.5 mm) in the anterior chamber due to poor injection or oil leaking, in which the IOP initially increased but dropped soon afterwards

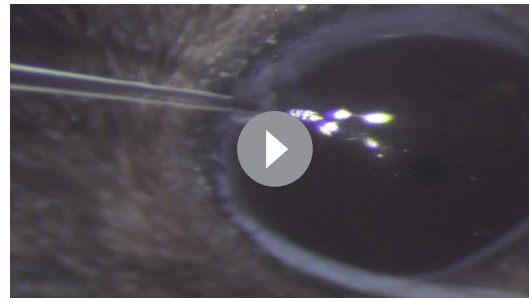

**Video 1.** Intracameral SO injection. Demonstration of the anterior chamber SO injection with a glass pippet and the SO droplet formation on top of iris to block pupil.

DOI: https://doi.org/10.7554/eLife.45881.004

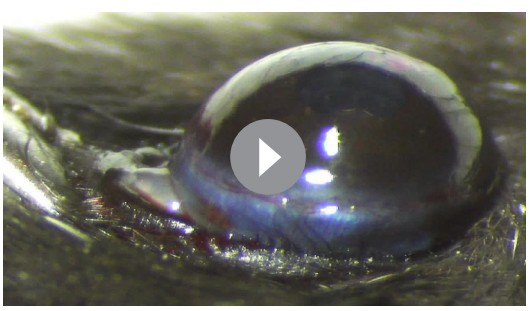

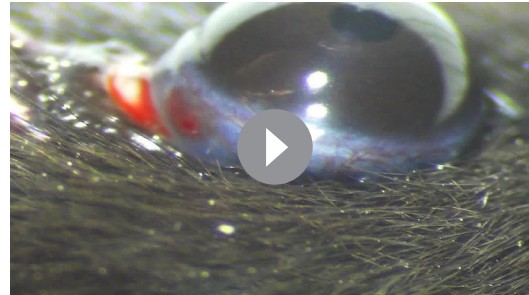

**Video 2.** Dye migration from vitreous chamber to anterior chamber in naïve eyes. DiI injected into the posterior chamber of the naïve eye and migrated into the anterior chamber.
DOI: https://doi.org/10.7554/eLife.45881.005

**Video 3.** Dye migration blocked in SOHU eyes. DiI injected into the posterior chamber of the SOHU eye and there was no DiI detected in the anterior chamber.
DOI: https://doi.org/10.7554/eLife.45881.006

(*Figure 1D*). Therefore, by observing the size of the SO droplet, it is convenient for us to identify mice very early that will not show elevated IOP and exclude them from subsequent experiments.

## Visual function deficits and dynamic morphological changes in SOHU eyes of living animals

To determine the dynamic changes in RGC morphology and function in SOHU eyes, we longitudinally measured the thickness of the ganglion cell complex (GCC) by OCT (*Nakano et al., 2011*), visual acuity by the optokinetic tracking response (OKR) (*Prusky et al., 2004*; *Douglas et al., 2005*), and general RGC function by pattern electroretinogram (PERG) (*Porciatti, 2015*) in living animals. Clinically, the thickness of the retinal nerve fiber layer (RNFL) measured by posterior OCT serves as a reliable biomarker for glaucomatous RGC degeneration (*Balcer et al., 2015*; *Aktas et al., 2016*; *Costello et al., 2006*). Because the mouse RNFL is too thin to be reliably measured, we used the thickness of GCC (*Nakano et al., 2011*), including RNFL, ganglion cell layer (GCL) and inner plexiform layer (IPL) together, to monitor degeneration of RGC axons, somata, and dendrites caused by ocular hypertension. GCC in SOHU eyes became gradually and progressively thinner (about 84%, 65%, 61% and 53% of CL eyes) at 1, 3, 5, and 8 weeks post injection (wpi). GCC thinning is statistically significant at 5 and 8 wpi compared to 1 wpi (*Figure 2A,B*). These results indicate progressive RGC degeneration in response to IOP elevation in SOHU eyes.

OKR is a natural reflex that objectively assesses mouse visual acuity (*Prusky et al., 2004*; *Douglas et al., 2005*). The mouse eye will only track a grating stimulus that is moving from the temporal to nasal visual field, which allows both eyes to be measured independently (*Douglas et al., 2005*; *Douglas et al., 2006*). It has been used to establish correlations between visual deficit and RGC loss in the DBA/2 glaucoma mouse model (*Burroughs et al., 2011*). The visual acuity of SOHU eyes decreased rapidly at 1wpi, which may due to the presence of SO in the anterior chamber. However, the further decreased visual acuity at 5 and 8 wpi compared to 1 wpi indicates progressive visual function deficits in the SOHU eyes (*Figure 2C*). PERG is an important electrophysiological assessment of general RGC function, in which the ERG responses are stimulated with contrast-reversing horizontal bars alternating at constant mean luminance (*Porciatti, 2015*). Our PERG system measured both eyes at the same time, so there was an internal control to use as a reference and normalization to minimize the variations. Consistent with visual acuity deficit, the P1-N2

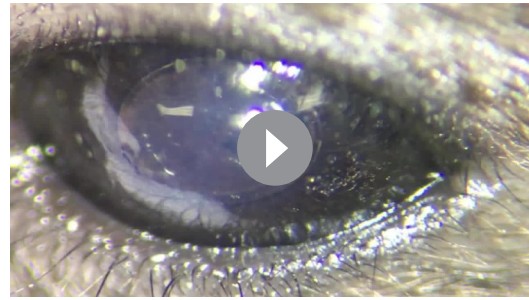

**Video 4.** SO droplet flows away from pupil after dilation. After pupil dilation, the SO droplet was pushed away from the pupil and iris by aqueous humor flooded into the anterior chamber.
DOI: https://doi.org/10.7554/eLife.45881.007

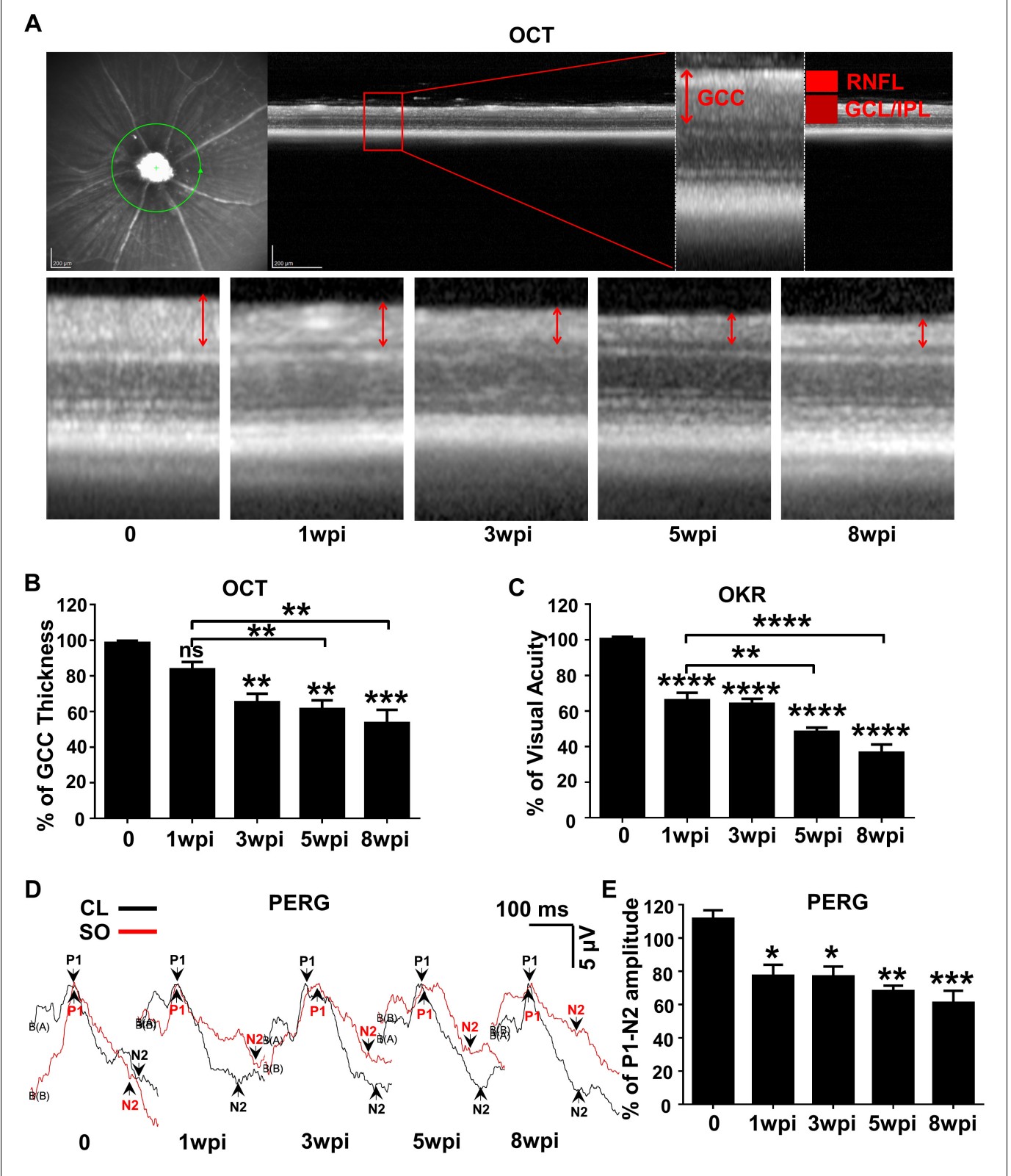

**Figure 2.** Dynamic changes in RGC morphology and visual function in living SOHU animals. (**A**) Representative OCT images of mouse retina. Green circle indicates the OCT scan area surrounding ON head. GCC: ganglion cell complex, including RNFL, GCL and IPL layers; indicated by double end arrows. (**B**) Quantification of GCC thickness, represented as percentage of GCC thickness in the SO eyes, compared to the CL eyes. n = 10–20. (**C**) Visual acuity measured by OKR, represented as percentage of visual acuity in the SO eyes, compared to the CL eyes. n = 10–20. (**D**) Representative

*Figure 2 continued on next page*

*Figure 2 continued*

waveforms of PERG in the contralateral control (CL, black lines) and the SO injected (SO, red lines) eyes at different time points after SO injection. P1: the first positive peak after the pattern stimulus; N2: the second negative peak after the pattern stimulus. (E) Quantification of P1-N2 amplitude, represented as percentage of P1-N2 amplitude in the SO eyes, compared to the CL eyes. $n$ = 13–15. Data are presented as means ± s.e.m, *: p<0.05, **: p<0.01, ***: p<0.001, ****: p<0.0001, one-way ANOVA with Tukey's multiple comparison test.

DOI: https://doi.org/10.7554/eLife.45881.008

amplitude ratio of the SO eyes to CL eyes decreased significantly (*Figure 2D,E*). However, that the lack of progression of PERG amplitude reduction suggests the SO itself may affect the light stimulation and PERG signal or the limitations of detection by PERG. Nevertheless, these results suggest that RGCs are very sensitive to IOP elevation, but resilient for a period of time before further degeneration. Taken together, these in vivo results show that SOHU eyes developed progressive structural and visual function deficits that closely resemble changes in glaucoma patients.

## Glaucomatous degeneration of RGC somata and axons in SOHU eyes

In vivo functional and imaging results indicate significant neurodegeneration in SOHU eyes, and histological analysis of post-mortem tissue samples supports these findings. We quantified surviving RGC somata in retinal wholemounts and surviving axons in ON semithin cross-sections at multiple time points after SO injection. Similar to the changes of GCC thickness measured by OCT in vivo, there was no statistical significance in surviving RGC counts in the peripheral retina between SOHU and control eyes at 1wpi, whereas there was significant and worsening RGC loss at 3, 5 and 8wpi, when only 43, 28, and 12% of peripheral RGCs survived (*Figure 3A,B*). This result confirmed significant progressive RGC death in response to IOP elevation in SOHU eyes. Significant RGC axon degeneration also occurred in SOHU ONs; only 57, 41% and 35% RGC axons survived at 3, 5, and 8wpi (*Figure 3A,C*). Therefore, IOP elevation in SOHU mouse eyes produces glaucomatous RGC and ON degeneration that starts as early as 3wpi and becomes progressing more severe at later time points that correlate with visual function deficits.

Although the SO used in these studies was sterile and safe for human use, we considered that toxicity might play a role in RGC death. Two experiments, however, provided evidence against this possibility: First, SO intravitreal injection did not cause significant IOP elevation, visual function deficits, or RGC/ON degeneration at 8wpi (*Figure 4A–F*). Second, the eyes with small SO droplets (≤1.5 mm) and unstable IOP elevation (*Figure 1D*) showed no significant RGC death or axon degeneration at 8wpi (*Figure 4G,H*). Therefore, we conclude that the neurodegeneration phenotypes observed in SOHU eyes are glaucomatous responses to ocular hypertension.

## SOHU is a reversible ocular hypertension model

One of the disadvantages of many other glaucoma models is that the initial eye injury is irreversible. However, we were able to flush out the SO from the anterior chamber with the aid of normal saline infiltration (*Figure 5A*, *Video 5*). This procedure lowered the IOP back to normal quickly and stably (*Figure 5B*), suggesting that SOHU is a reversible model that can be used to test whether lowering IOP affects degeneration of glaucomatous RGCs or the combination effect with neuroprotection.

## Discussion

A reliable animal glaucoma model that closely mimics the disease in humans is a prerequisite for studies of pathogenetic mechanisms and for selecting efficient neuroprotective treatments for clinical use. In the present study, we developed a highly effective and reproducible method adopted from a clinical secondary glaucoma complication after retina surgery. Injection of SO to the mouse anterior chamber efficiently induces a series of reactions, including pupillary block, blockage of the aqueous humor outflow from anterior chamber, accumulation of aqueous humor in the posterior chamber, closure of the anterior chamber angle, and IOP elevation. These reactions occur without causing overt ocular structural damage or inflammatory responses while simulating acute glaucomatous changes that human patients develop over years by inducing progressive RGC and ON degeneration and visual functional deficits within weeks.

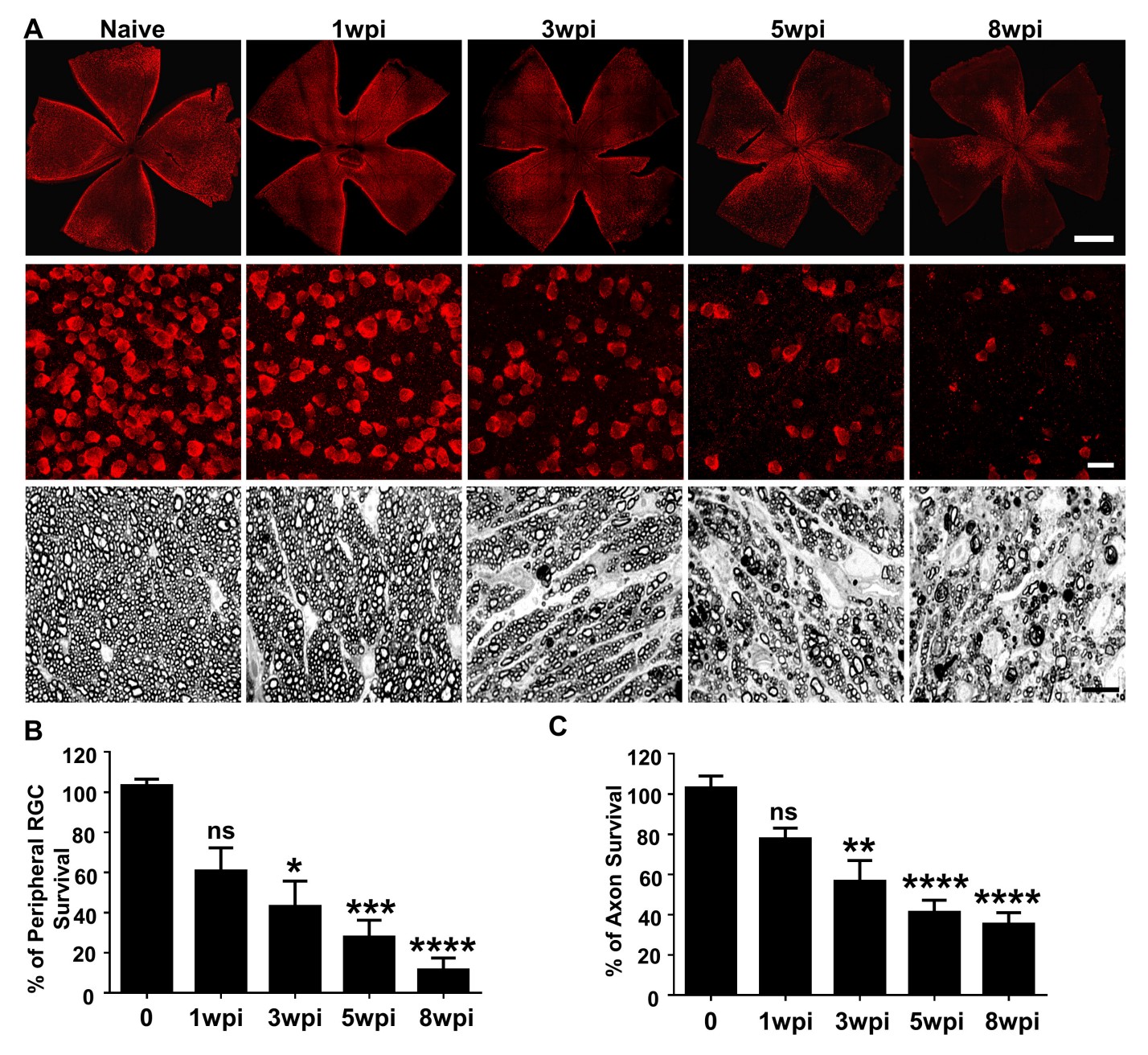

**Figure 3.** Glaucomatous RGC soma and axon degeneration in SOHU eyes. (**A**) Upper panel, confocal images of whole flat-mounted retinas showing surviving RBPMS-positive (red) RGCs at different time points after SO injection. Scale bar, 100 µm. Middle panel, confocal images of a portion of flat-mounted retinas showing surviving RBPMS-positive (red) RGCs at different time points after SO injection. Scale bar, 20 µm. Lower panel, light microscope images of semi-thin transverse sections of ON stained with PPD at different time points after SO injection. Scale bar, 10 µm. (**B,C**) Quantification of surviving RGCs in the peripheral retina (*n* = 11–13) and surviving axons in ON (*n* = 10–16) at different time points after SO injection, represented as percentage of SO eyes compared to CL eyes. Data are presented as means ± s.e.m. *p<0.05, **p<0.01, ***: p<0.001, ****: p<0.0001; one-way ANOVA with Tukey's multiple comparison test.

DOI: https://doi.org/10.7554/eLife.45881.009

SO injection is limited to one eye in each mouse, with the other eye receiving an equivalent volume of normal saline. This serves as a convenient internal control for the surgical procedure and for studies of RGC morphology and function. It is reasonable to conclude that IOP is elevated in the SOHU eyes because of impeded inflow and accumulation of aqueous humor in the posterior

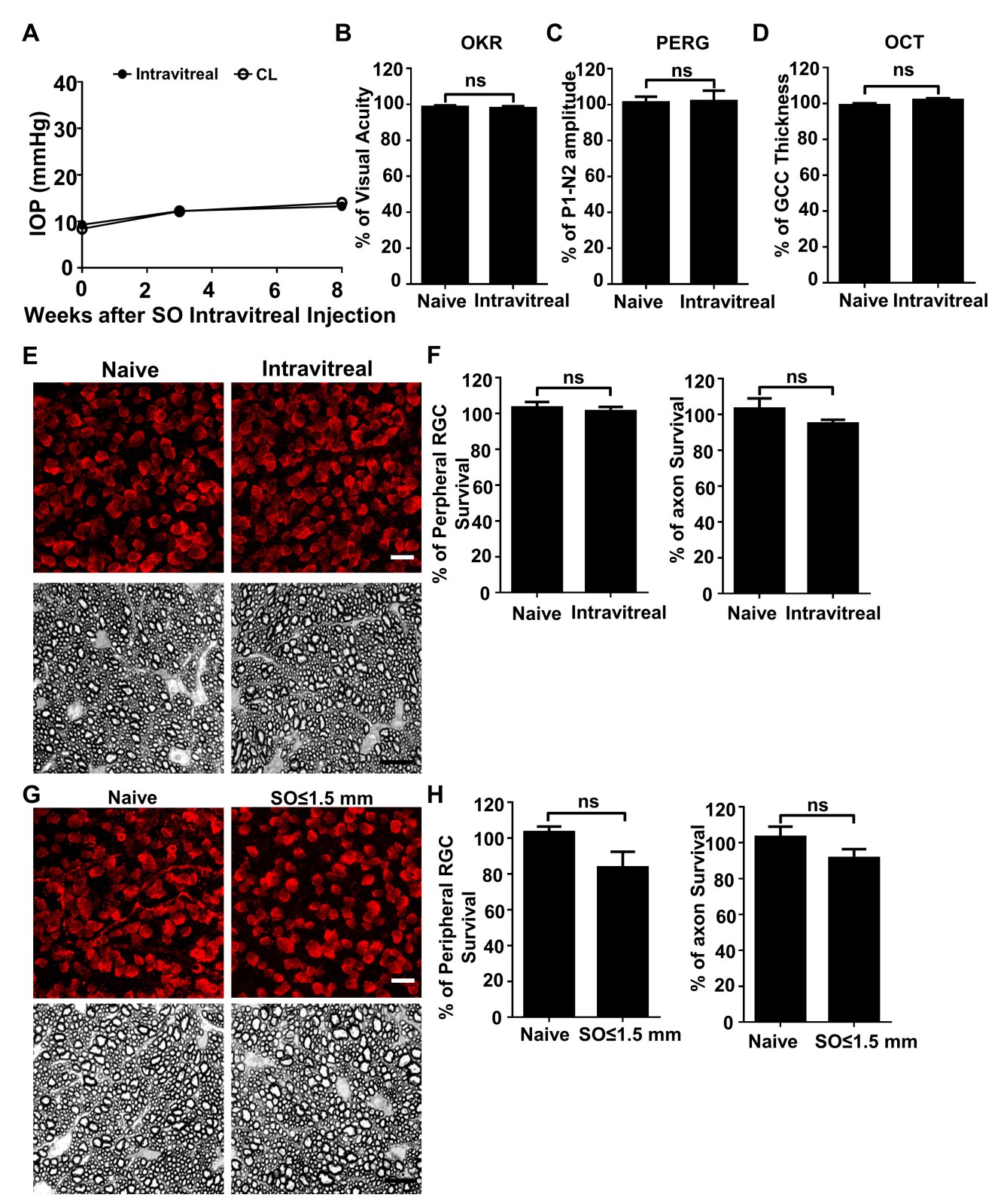

**Figure 4.** SO itself does not cause glaucomatous degeneration. (A) IOP measurements at different time points after intravitreal SO injection. n = 15. (B) Visual acuity measured by OKR, represented as percentage of visual acuity in the SO eyes, compared to the CL eyes. n = 13–15. (C) Quantification of P1-N2 amplitude of PERG, represented as percentage of P1-N2 amplitude in the SO eyes, compared to the CL eyes. n = 12–15. (D) Quantification of GCC thickness measured by OCT, represented as percentage of GCC thickness in the SO eyes, compared to the CL eyes. n = 11–13. (E) Upper panel,
*Figure 4 continued on next page*

*Figure 4 continued*

confocal images of portions of flat-mounted retinas showing surviving RBPMS-positive (red) RGCs at 8wpi after intravitreal SO injection and contralateral naive eye. Scale bar, 20 µm. Lower panel, light microscope images of semi-thin transverse sections of ON stained with PPD at 8wpi after intravitreal SO injection and contralateral naive eye. Scale bar, 10 µm. (F) Quantification of surviving RGCs (n = 10) and surviving axons in ON (n = 10) at 8wpi after intravitreal SO injection, represented as percentage of SO eyes compared to the CL eyes. Data are presented as means ± s.e.m, Student t-test. (G) Upper panel, confocal images of portion of flat-mounted retinas showing surviving RBPMS positive (red) RGCs at 8wpi after intracameral SO injection (small size of SO droplet,≤1.5 mm) and contralateral naive eye. Scale bar, 20 µm. Lower panel, light microscope images of semi-thin transverse sections of ON stained with PPD at 8wpi after intracameral SO injection and contralateral naive eye. Scale bar, 10 µm. (H) Quantification of surviving RGCs (n = 12) and surviving axons in ON (n = 13) at 8wpi, represented as percentage of SO eyes compared to the CL eyes. Data are presented as means ± s.e.m, Student t-test.

DOI: https://doi.org/10.7554/eLife.45881.010

segment of the eye, rather than by an aspect of the surgical procedure, such as the cornea wound or inflammation, which was rare. Although we never observed small emulsified SO droplets in any of

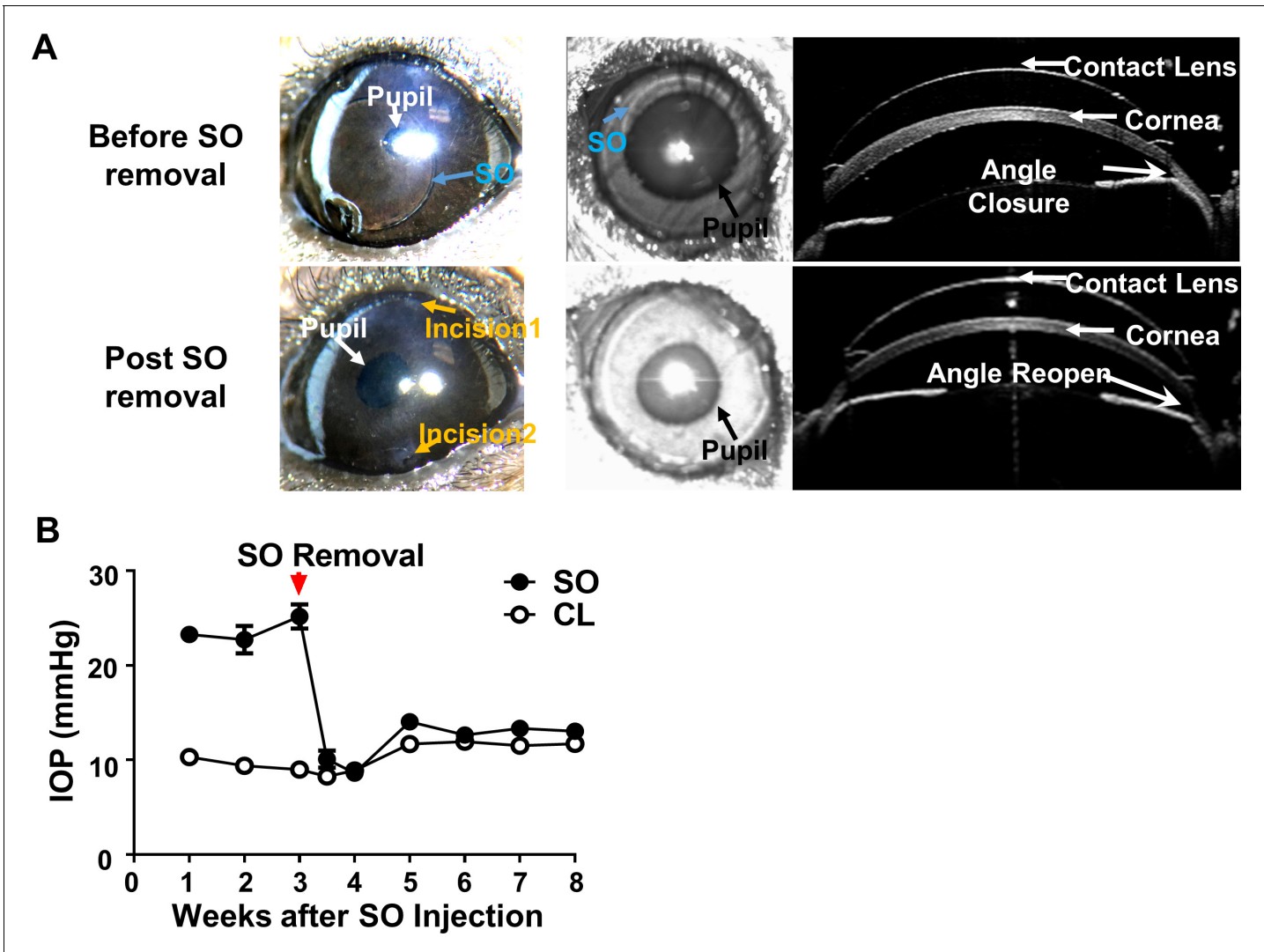

**Figure 5.** SOHU is reversible by SO removal. (A) Representative images of SOHU eyes before and after SO removal, and anterior chamber OCT images in living animals showing the relative size of SO droplet to pupil and the corresponding closure or opening of the anterior chamber angle before and after SO removal. (B) IOP measurements before and after SO removal at different time points. n = 16.

DOI: https://doi.org/10.7554/eLife.45881.011

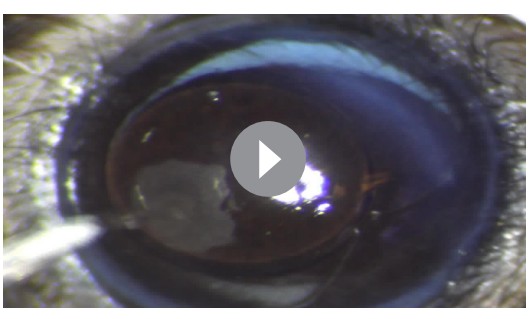

**Video 5.** SO removal from SOHU eyes. To remove SO from the anterior chamber, one needle is used to flush normal saline into the anterior chamber from one side of the cornea and another glass pippet was used to suck away the SO from the anterior chamber.
DOI: https://doi.org/10.7554/eLife.45881.012

the mouse eyes, we cannot exclude the possibility that at least in some cases oil occluded the TM. One previous glaucoma model elevated IOP in rats by injecting hyaluronic acid to impede aqueous outflow from TM (*Mayordomo-Febrer et al., 2015*; *Benozzi et al., 2002*; *Moreno et al., 2005*), indicating the possibility of TM damage due to repeated injection of a viscoelastic solution into anterior chamber. However, two of our observations provide evidence against this notion by indicating that TM function is normal in our model: 1. Pupillary dilation for an extended period of time eventually allows adequate aqueous clearance through TM and downregulation of IOP. 2. SO removal quickly returns IOP to normal, which indicates that the SO droplet is a prerequisite for IOP elevation. The relatively small variability in the duration and magnitude of IOP elevation in SOHU eyes after a single injection makes it a simple and reliable ocular hypertension model, which can be explained by the persistence of a SO droplet that is large relative to the size of the pupil.

Because of the unique feature of pupillary block associated with SOHU, the IOP is elevated in the posterior part of the eye, but not in the anterior chamber. We postulate that, after the pupil is sealed by SO, the large mouse lens, together with the iris and ciliary body, forms a rigid barrier that essentially disconnects the anterior and posterior chambers and thus shields the anterior chamber from the high pressure in the posterior chamber. This pathogenesis gives the model two advantageous characteristics: 1) The anterior segments of the experimental eyes are not substantially affected, leaving clear ocular elements that allow easy and reliable assessment of in vivo visual function and morphology; 2) The high IOP of the posterior chamber causes pronounced glaucomatous neurodegeneration within 5–8 weeks, which facilitates testing neuroprotectants by allowing any benefit to be detected in a short period of experimental time. One caveat, however, is that SO itself in anterior chamber may blur vision or affect the visual function assays because its optical characteristics differ from those of aqueous humor. These differences may cause early decreases in visual acuity and PERG amplitude at 1wpi, when OCT imaging, which does not depend on the transparency of anterior segment of the eye, shows no significant morphological degeneration. It is also possible that deficits in visual function precede morphological changes, or that there is no proportional relationship between RGC function and RGC morphology, since the visual acuity and PERG amplitude are not always correlated with RGC numbers. An assay of visual function that is unaffected by SO in the anterior chamber and that is more quantitatively related to RGC numbers is needed to resolve the discrepancy definitively.

The SOHU model is excellent for deciphering the key components of the degeneration cascade associated with ocular hypertension, but it is not suitable for TM function/deficit studies because it depends on pupillary block and spares TM. Because the IOP elevation is rapid and neurodegeneration severe within a few weeks, the SOHU model has features of acute secondary glaucoma in humans, but the extent to which it also mimics more chronic and milder primary glaucoma in patients needs further investigation. Human secondary angle-closure glaucoma is accompanied by RGC and significant photoreceptor loss (*Panda and Jonas, 1992*; *Janssen et al., 1996*; *Nork et al., 2000*), which may at least in part be due to ischemia caused by high IOP. Decreased blood flow through the choroidal circulation and ophthalmic artery has been reported in primary open angle glaucoma patients as well (*Rojanapongpun et al., 1993*; *Michelson et al., 1995*; *Cellini et al., 1996*; *Yamazaki and Drance, 1997*; *Butt et al., 1997*; *Kaiser et al., 1997*; *Yin et al., 1997*). A modified SOHU model that induces and maintains a moderate elevation of IOP through frequent pupil dilation may more closely reproduce the features of clinical primary open angle glaucoma.

In summary, this novel mouse acute ocular hypertension glaucoma model replicates secondary post-operative glaucoma. It is straightforward, does not require special equipment or repeat injections, and may be applicable to a range of animal species with only minor modifications. It is easily

reversible by removing SO from the anterior chamber and particularly useful for screening neuroprotective therapies in vivo. Therefore we report this simple, convenient, effective, reproducible, and reversible mouse model that generates stable, robust IOP elevation and significant neurodegeneration within weeks with the hopes that it will standardize assessment of the pathogenesis of ocular hypertension-induced glaucoma and facilitate selection of neuroprotectants for glaucoma.

# Materials and methods

**Key resources table**

| Reagent type (species) or resource | Designation | Source or reference | Identifiers | Additional information |
|---|---|---|---|---|
| Strain, strain background (Mus musculus) | C57BL/6J | Jackson Laboratories | 000664 | |
| Antibody | anti-RBPMS (guinea pig polyclonal) | Custom-made by ProSci | | 1:4000 |
| Antibody | Cy3 Goat anti-Guinea Pig IgG | Jackson ImmunoResearch | 106-165-003 | 1:200 |
| Chemical compound, drug | Silicone oil | Alcon Laboratories | 1,000 mPa.s, Silikon | |
| Software, algorithm | Graphpad prism6 | GraphPad Software | | |
| Software, algorithm | Volocity software | Quorum Technologies | | |

## Mice

C57BL/6J WT mice were purchased from Jackson Laboratories (Bar Harbor, Maine). For all surgical and treatment comparisons, control and treatment groups were prepared together in single cohorts, and the experiment repeated at least twice. All experimental procedures were performed in compliance with animal protocols approved by the IACUC at Stanford University School of Medicine.

## Induction of IOP elevation by intracameral injection of SO

Male mice received SO injection at 9–10 weeks of age. Mice were anesthetized by an intraperitoneal injection of Avertin (0.3 mg/g) instead of ketamine/xylazine to avoid pupil dilation. The mice were then placed in a lateral position on a surgery platform. Prior to injection, one drop of 0.5% proparacaine hydrochloride (Akorn, Somerset, New Jersey) was applied to the cornea to reduce its sensitivity during the procedure. A 32G needle was tunneled through the layers of the cornea at the superotemporal side close to the limbus to reach the anterior chamber without injuring lens or iris. Following this entry, about 2 µl silicone oil (1,000 mPa.s, Silikon, Alcon Laboratories, Fort Worth, Texas) were injected slowly into the anterior chamber using a homemade sterile glass micropipette, until the oil droplet expanded to cover most areas of the iris. The micropipette was held in place for 30 s before withdrawing it slowly. After the injection, the upper eyelid was gently massaged to close the corneal incision to minimize SO leakage, and veterinary antibiotic ointment (BNP ophthalmic ointment, Vetropolycin, Dechra, Overland Park, Kansas) was applied to the surface of the injected eye. The contralateral control eyes received 2 µl normal saline to the anterior chamber. During the whole procedure, artificial tears (Systane Ultra Lubricant Eye Drops, Alcon Laboratories, Fort Worth, Texas) were applied to keep the cornea moist. The rare mouse that showed corneal opacity associated with band-shaped degeneration or neovascularization was excluded from further analysis.

## Removing SO from the anterior chamber

The oil droplet was removed from the anterior chamber at 3wpi. Mice were anesthetized by intraperitoneal injection of Avertin (0.3 mg/g) and placed in a lateral position on a surgery platform. Prior to injection, one drop of 0.5% proparacaine hydrochloride (Akorn, Somerset, New Jersey) was applied to the cornea to reduce its sensitivity during the procedure. Then two corneal tunnel

incisions were made using a 32G needle: one tunnel incision superior and one tunnel incision inferior to the center of the cornea, each at the edge of the oil droplet. A 33G needle attached to an elevated balanced salt solution plus (BSS Plus, Alcon Laboratories, Ft. Worth, Texas) drip (110 cm $H_2O$ height, equal to 81 mmHg) was inserted through the superior corneal incision to flow BSS into anterior chamber to maintain its volume. At the same time, another 33G needle attached to a 1 mL syringe with the plunger removed, was inserted through the inferior tunnel incision to allow SO outflow. After removing the oil, a small air bubble was injected by a glass micropipette into anterior chamber to maintain the volume of anterior chamber and temporarily seal the corneal incision. Veterinary antibiotic ointment (BNP ophthalmic ointment) was applied to the surface of the eye.

## IOP measurement

The IOP of both eyes was monitored once weekly until 8 weeks after SO injection using the TonoLab tonometer (Colonial Medical Supply, Espoo, Finland) according to product instructions. Briefly, in the morning, mice were anesthetized with a sustained flow of isoflurane (3% isoflurane at 2 L/minute mixed with oxygen) delivered to the nose by a special rodent nose cone (Xenotec, Inc, Rolla, Missouri), which left the eyes exposed for IOP measurement. The TonoLab tonometer takes five measurements, eliminates high and low readings and generates an average. We considered this machine-generated average as one reading. Three machine-generated readings were obtained from each eye every 5 min, and the mean was calculated to determine the IOP. During this procedure, artificial tears were applied to keep the cornea moist.

## Immunohistochemistry of whole-mount retina and RGC counting

After transcardiac perfusion with 4% PFA in PBS, the eyes were dissected out, post-fixed with 4% PFA for 2 hr, at room temperature, and cryoprotected in 30% sucrose at 4°C overnight. Retinas were dissected out and washed extensively in PBS before blocking in staining buffer (10% normal goat serum and 2% Triton X-100 in PBS) for half an hour. RBPMS guinea pig antibody made at ProSci, California according to publications (*Kwong et al., 2010*; *Rodriguez et al., 2014*) and used at 1:4000, and rat HA (clone 3F10, 1:200, Roche) were diluted in the same staining buffer. Floating retinas were incubated with primary antibodies overnight at 4°C and washed 3 times for 30 min each with PBS. Secondary antibodies (Cy2 or Cy3) were then applied (1:200–400; Jackson ImmunoResearch, West Grove, Pennsylvania) and incubated for 1 hr at room temperature. Retinas were again washed 3 times for 30 min each with PBS before a cover slip was attached with Fluoromount-G (SouthernBiotech, Birmingham, Alabama). For peripheral RGC counting, whole-mount retinas were immunostained with the RBPMS antibody, 6–9 fields sampled from peripheral regions of each retina using 40x lens with a Zeiss M2 epifluorescence microscope, and RBPMS +RGCs counted by Velocity software (Quorum Technologies). The percentage of RGC survival was calculated as the ratio of surviving RGC numbers in injured eyes compared to contralateral uninjured eyes. The investigators who counted the cells were masked to the treatment of the samples.

## ON semi-thin sections and quantification of surviving axons

After mice were perfused through the heart with ice cold 4% paraformaldehyde (PFA) in PBS, the ON was exposed by removing the brain and post-fixed in situ using 2% glutaraldehyde/2% PFA in 0.1M PB for 4 hr on ice. Samples were then washed with 0.1M PB three times, 10 min each wash. The ONs were then carefully dissected out and rinsed with 0.1M PB three times, 10 min each wash. They were then incubated in 1% osmium tetroxide in 0.1M PB for 1 hr at room temperature followed by washing with 0.1M PB for 10 min and water for 5 min. ONs were next dehydrated through a series of graded ethanol (50% to 100%), rinsed twice with propylene oxide (P.O.), 3 min each rinse, and transferred to medium containing 50% EMbed 812/50% P.O. overnight. The next day, the medium was changed to a 2:1 ratio of EMbed 812/P.O. ONs remained in this mixture overnight, then were transferred to 100% EMbed 812 on a rotator for another 6 hr, embedded in a mold filled with 100% EMbed 812 and incubated at 60°C overnight. Semi-thin sections (1 μm) were cut on an ultramicrotome (EM UC7, Leica, Wetzlar, Germany) and collected 2 mm distal to the eye. The semi-thin sections were attached to glass slides and stained with 1% para-phenylenediamine (PPD) in methanol: isopropanol (1:1) for 35 min. After rinsing three times with methanol: isopropanol (1:1), coverslips were applied with Permount Mounting Medium (Electron Microscopy Sciences, Hatfield,

Pennsylvania). PPD stains all myelin sheaths, but darkly stains the axoplasm only of degenerating axons, which allows us to differentiate surviving axons from degenerating axons (*Smith et al., 2002*). Four sections of each ON were imaged through a 100x lens of a Zeiss M2 epifluorescence microscope to cover the entire area of the ON without overlap. Two areas of 21.4 μm X 29.1 μm were cropped from the center of each image, and the surviving axons within the designated areas were counted manually. After counting all the images taken from a single nerve, the mean of the surviving axon number was calculated for each ON. The mean of the surviving axon number in the injured ON was compared to that in the contralateral control ON to yield a percentage of axon survival value. The investigators who counted the axons were masked to the treatment of the samples.

### Intravitreal injection

These procedures have been described previously (*Hu et al., 2012*; *Yang et al., 2014*). Briefly, mice were anesthetized by xylazine and ketamine based on their body weight (0.01 mg xylazine/g + 0.08 mg ketamine/g). For intravitreal dye injection, DiI solution (ThermoFisher Scientific, V22885) was injected into posterior chamber through the point directly behind the limbus (beneath the iris) to demonstrate aqueous humor migration.

### Pattern electroretinogram (PERG) recording

Mice were anesthetized by xylazine and ketamine based on their body weight (0.01 mg xylazine/g + 0.08 mg ketamine/g). PERG recording of both eyes was performed at the same time with the Miami PERG system (Intelligent Hearing Systems, Miami, FL) according to published protocol (*Chou et al., 2014*). Briefly, mice were placed on a feedback-controlled heating pad (TCAT-2LV, Physitemp Instruments Inc, Clifton, New Jersey) to maintain animal core temperature at 37˚C. A small lubricant eye drop (Systane Ultra Lubricant Eye Drops, Alcon Laboratories, Ft. Worth, Texas) was applied before recording to prevent corneal dryness. The reference electrode was placed subcutaneously on the back of the head between the two ears and the ground electrode was placed at the root of the tail. The active steel needle electrode was placed subcutaneously on the snout for the simultaneous acquisition of left and right eye responses. Two 14 cm x 14 cm LED-based stimulators were placed in front so that the center of each screen was 10 cm from each eye. The pattern remained at a contrast of 85% and a luminance of 800 cd/m$^2$, and consisted of four cycles of black-gray elements, with a spatial frequency of 0.052 c/d. Upon stimulation, the independent PERG signals were recorded from the snout and simultaneously by asynchronous binocular acquisition. With each trace recording up to 1020 ms, two consecutive recordings of 200 traces were averaged to achieve one readout. The first positive peak in the waveform was designated as P1 (typically around 100 ms) and the second negative peak as N2 (typically around 205 ms). The amplitude was measured from P1 to N2. The mean of the P1-N2 amplitude in the injured eye was compared to that in the contralateral control eye to yield a percentage of amplitude change. The investigators who measured the amplitudes were masked to the treatment of the samples.

### Spectral-domain optical coherence tomography (SD-OCT) imaging

After the mice were anesthetized, pupils were dilated by applying 1% tropicamide sterile ophthalmic solution (Akorn, Somerset, New Jersey), and a customized +10D contact lens (3.0 mm diameter, 1.6 mm BC, PMMA clear, Advanced Vision Technologies) applied to the dilated pupil. The retina fundus images were captured with the Heidelberg Spectralis SLO/OCT system (Heidelberg Engineering, Germany) equipped with an 870 nm infrared wavelength light source and a 30° lens (Heidelberg Engineering). The OCT scanner has 7 μm optical axial resolution, 3.5 μm digital resolution, and 1.8 mm scan depth at 40 kHz scan rate. The mouse retina was scanned with the ring scan mode centered by the optic nerve head at 100 frames average under high-resolution mode (each B-scan consisted of 1536 A scans). The GCC includes retinal nerve fiber layer (RNFL), ganglion cell layer (GCL) and inner plexiform layer (IPL). The average thickness of GCC around the optic nerve head was measured manually with the aid of Heidelberg software. The mean of the GCC thickness in the injured retina was compared to that in the contralateral control retina to yield a percentage of GCC thickness value. The investigators who measured the thickness of GCC were masked to the treatment of the samples.

## OKR measurement

To measure the spatial vision using the opto-kinetic response (OKR), mice were placed unrestrained on a platform in the center of four 17-inch LCD computer monitors (Dell, Phoenix, AZ), with a video camera above the platform to capture the movement of the mouse. A rotating cylinder with vertical sine wave grating was computed and projected to the four monitors by OptoMotry software (CerebralMechanics Inc, Lethbridge, Alberta, Canada). The sine wave grating, consisting of black (mean luminance 0.22 cd/m2) and white (mean luminance 152.13 cd/m2) at 100% contrast and 12 degree/second, provides a virtual-reality environment to measure the spatial acuity of left eye when rotates clockwise and right eye when it rotates counterclockwise. Initially, the monitors were covered with gray so that the mouse calmed down and stopped moving, then the gray was switched to a low spatial frequency (0.1 cycle/degree) for five seconds, during which the mouse was assessed for whether the head turned to track the grating. The short time frame of assessment ensures that the mice did not adapt to the stimulus, which would lead to false readouts. When the mouse was determined to be capable of tracking the grating, the spatial frequency was increased repeatedly until the maximum frequency was identified and recorded. At each time point, the maximum frequency of the experimental eye was compared to that of the contralateral eye. The mice were tested in the morning and the investigator who judged the OKR was masked to the treatment of mice.

## Statistical analyses

GraphPad Prism six was used to generate graphs and for statistical analyses. Data are presented as means ± s.e.m. Student's t-test was used for two groups comparison and One-way ANOVA with post hoc test was used for multiple comparisons.

## Acknowledgements

We thank Drs. Zhigang He and Alan Tessler for critically reading the manuscript. We also appreciate the help from Gang Jiang and Niannian Liu in making schematic diagrams in *Figure 1A*. YH is supported by NIH grants EY024932, EY023295 and EY028106 and grants from BrightFocus Foundation, Glaucoma Research Foundation, National Multiple Sclerosis Society and William and Mary Greve Special Scholar Award from Research to Prevent Blindness. Portions of this work were supported by NIH grants EY026766 and EY027261 to JLG and NIH grants EY-25295, K08-EY022058, VA CX001298, Ziegler Foundation for the Blind to YS, who is a Stanford Child Health Research Institute Laurie Kraus Lacob Faculty Scholar. HCW is supported by NIH T32 Postdoctoral Fellowship (NEI T32 EY027816). We are grateful for an unrestricted grant from Research to Prevent Blindness and NEI P30-026877 to the Department of Ophthalmology. The authors have declared that no conflict of interest exists.

## Additional information

### Funding

| Funder | Grant reference number | Author |
|---|---|---|
| National Eye Institute | NEI T32 EY027816 | Hannah C Webber |
| National Eye Institute | EY-25295 | Yang Sun |
| National Eye Institute | K08-EY022058 | Yang Sun |
| U.S. Department of Veterans Affairs | VA CX001298 | Yang Sun |
| E. Matilda Ziegler Foundation for the Blind | | Yang Sun |
| National Eye Institute | EY026766 | Jeffrey L Goldberg |
| National Eye Institute | EY027261 | Jeffrey L Goldberg |
| National Eye Institute | EY024932 | Yang Hu |
| National Eye Institute | EY023295 | Yang Hu |

| National Eye Institute | EY028106 | Yang Hu |
|---|---|---|
| BrightFocus Foundation | | Yang Hu |
| Glaucoma Research Foundation | | Yang Hu |
| National Multiple Sclerosis Society | | Yang Hu |
| Research to Prevent Blindness | William and Mary Greve Special Scholar Award | Yang Hu |

The funders had no role in study design, data collection and interpretation, or the decision to submit the work for publication.

## Author contributions

Jie Zhang, Liang Li, Conceptualization, Data curation, Formal analysis, Validation, Investigation, Visualization, Methodology, Writing—review and editing; Haoliang Huang, Data curation, Formal analysis, Validation, Investigation, Visualization, Methodology, Writing—review and editing; Fang Fang, Data curation, Validation, Methodology; Hannah C Webber, Investigation, Visualization, Writing—review and editing; Pei Zhuang, Formal analysis, Validation, Investigation, Visualization, Methodology, Writing—review and editing; Liang Liu, Validation, Investigation, Methodology; Roopa Dalal, Investigation, Methodology; Peter H Tang, Vinit B Mahajan, Shaohua Li, Mingchang Zhang, Conceptualization, Writing—review and editing; Yang Sun, Investigation, Writing—review and editing; Jeffrey L Goldberg, Conceptualization, Investigation, Writing—review and editing; Yang Hu, Conceptualization, Resources, Data curation, Formal analysis, Supervision, Funding acquisition, Investigation, Methodology, Writing—original draft, Project administration, Writing—review and editing

## Author ORCIDs

Jeffrey L Goldberg (iD) http://orcid.org/0000-0002-1390-7360
Yang Hu (iD) https://orcid.org/0000-0002-7980-1649

## Ethics

Animal experimentation: This study was performed in strict accordance with the recommendations in the Guide for the Care and Use of Laboratory Animals of the National Institutes of Health. All of the animals were handled according to approved institutional animal care and use committee (IACUC) protocols (#32093) of the Stanford University.

## Decision letter and Author response

Decision letter https://doi.org/10.7554/eLife.45881.014
Author response https://doi.org/10.7554/eLife.45881.015

# Additional files

## Data availability

All data generated or analysed during this study are included in the manuscript and supporting files. Source data files have been provided for all the figures.

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
