## [Decision Letter]

[Editors’ note: this article was originally rejected after discussions between the reviewers, but the authors were invited to resubmit after an appeal against the decision.]

Thank you for submitting your work entitled "Silicone oil-induced ocular hypertension in mouse models glaucomatous neurodegeneration and neuroprotection" for consideration by *eLife*. Your article has been reviewed by three peer reviewers, and the evaluation has been overseen by a Reviewing Editor and a Senior Editor. The following individual involved in review of your submission has agreed to reveal their identity: Yvonne Ou (Reviewer #1).

While the reviewers found the work interesting, the number of substantive questions raised was such that we feel we must reject it. We hope that the reviewers' comments will be useful to you. We apologize for not being able to deliver better news, and we hope that you will continue to consider *eLife* for future submissions.

*Reviewer #1:*

The authors of this manuscript aim to establish a novel model of secondary glaucoma that is easily reproducible, develops quickly, and does not produce irreversible ocular tissue damage and inflammation, in which mechanism studies and drug screening can be easily performed. This model relies on the use of silicone oil to produce pupillary block, blockage of the aqueous humor outflow from anterior chamber, accumulation of aqueous humor in the posterior chamber, closure of the anterior chamber angle, and IOP elevation. Morphologically, there was a significant loss of RGC somata and axons and reduction in GCC starting at 3 wpi and progressively worsened. Functional assessment by OKR and PERG demonstrated a reduction in visual acuity and RGC function, respectively, as early as 1 wpi that progressively worsened, although this data needs to be clarified.

This model differs from other established models of experimental glaucoma, in that it has a high success rate of IOP elevation that can be easily predicted by SO diameter, severe cell loss, and minimal tissue damage and inflammation. Due to its ease of induction, rapid and severe development of glaucomatous neurodegeneration, this model will be valuable in investigating molecular mechanisms and screening drug compounds.

- What implications does mechanisms of secondary pupillary block angle-closure glaucoma have on primary open angle glaucoma? It would be worth mentioning that this type of postoperative secondary glaucoma is rare since surgeons prophylactically create peripheral iridotomies. Indeed, secondary glaucoma caused by SO is more frequently the result of emulsification of SO leading to very small bubbles lodged in the angle. Discussion, fourth paragraph, the last sentence is a bit overstated, and a clearer discussion of how this model differs from human glaucoma and POAG would be appreciated.

- How does the effect of IOP elevation in the posterior chamber differ from the anterior chamber? We had difficulty understanding the naming of the "SOHU" model. Is there known undermeasurement of IOP when the anterior chamber is filled with silicone oil?

- A more detailed explanation in the Materials and methods for IOP measurement would be appreciated. Was pupillary block removed before measuring IOP (i.e. eyes were dilated?)? Was IOP measured at the same time of day? The authors are trying to establish this model as a model of increased IOP that leads to neurodegenerative changes. IOP measurements fluctuate daily and reporting IOP measurements only once a week seems insufficient. Do light/dark cycles affect IOP elevation since pupillary block is relieved upon dilation that would occur in dark?

- Are OKR and PERG reliable functional assessments if visual acuity is impaired due to difference in refractive index of SO? Is the baseline "0" (where data is normalized to 100%) a measurement made at baseline in which neither eye has been perturbed, or is it measured right after SO and saline injections? The latter would be the correct control.

- The manner in which the data is presented for OCT, OKR, and PERG in Figure 2 do not show that there is progression over time. This data needs to be presented more clearly, since in the text the authors argue that there is progressive thinning (OCT), progressive loss of vision (OKR), and progressive decrease in the P1-N2 amplitude (PERG) with time. If there was no progression for OKR and PERG, then it suggests that the SO bubble itself is the cause for the decreased amplitudes.

- Figure 3, middle panel. What portion of the flat mount retina is represented (central/peripheral)? The authors should comment on why the peripheral retina appears to have much more significant loss than centrally. Also, how were the 6-9 fields "randomly" selected to quantify RBPMS? It seems that whether one chose fields from the center vs. periphery would play a large role in the quantification.

*Reviewer #2:*

This manuscript reports the development of a new method of inducing IOP elevation in mice by intracameral injection of silicone oil (SO). It has provided data to show that SO injection caused reduced visual acuity and GCC thickness, deficits in PERG, and the loss of retinal ganglion cells and axons. Modifying ER stress pathways by AAV delivery of CHOP shRNA and XBP-1s significantly prevented RGC and axon loss. It also showed that flushing out the SO by anterior chamber injection of saline allows the IOP level returning to normal. The model of SO-induced ocular hypertension is novel and can be very interesting. However, the manuscript presents several issues that need to be addressed:

1) TanoLab assessment for IOP in this model may not reflect the real IOP value in the back of the eye. As it is stated in the manuscript: "Because of the unique feature of pupillary block associated with SOHU, the IOP is elevated in the posterior part of the eye, but not in the anterior chamber". TanoLab measures the pressure of the anterior chamber, thus is incapable of assessing the actual IOP value in the posterior part of the eye, which is likely to be far higher than it was detected by TanoLab in this case.

2) The rate of RGC loss in this model is excessive. As data showed in Figure 3, elevation of IOP led to nearly 90% RGC loss (12% RGC survival) in 8 weeks; this represents a very severe damage, similar to the situation of complete transection of the optic nerve, and it does not replicate the most cases of glaucoma.

3) The manuscript showed either a large neuroprotective effect by control AAV injection (as suggested by the authors) or inconsistent neural damage from time to time in the SOHU model. In Figure 3, injection of SO ended up with 12% RGC survival by 8 weeks, while in Figure 4, counts of survival RGCs were ~60% by 8 weeks. Such a huge neuroprotective effect by control AAV injection has not been observed or reported by others. If the authors claimed that they found it in other optic neuropathy models, the data need to be provided.

*Reviewer #3:*

The proposal that silicone oil injection may be a useful mouse high pressure model is reasonable, and if it is true that the cornea stays clear, this would be an advantage over other mouse models. However, every model in mice shows corneal enlargement (and axial length increase) with IOP in the range reported here, so further studies of clarity need to be presented. Removal of the oil to lower IOP is potentially useful adjunct, if one wishes to study various periods of IOP elevation, then allow time for further events at normal IOP. Unfortunately, the work as presented does not adequately show the true primary outcomes definitively and objectively.

1) "Glaucoma is a neurodegenerative disease characterized by optic neuropathy with thinning of the retinal nerve fiber layer (RNFL) followed by progressive retinal ganglion cell (RGC) degeneration (Quigley, 1993; Quigley et al.,1995; Livvy et al., 2005; Howell et al., 2007; Weinreb and Khaw, 2004; Calkins, 2012; Burgoyne, 2011; Nickells et al., 2012; Jonas et al., 2017)." This is an incorrect and inexact definition of glaucomatous optic neuropathy. Glaucoma is characterized by injury to the axons of RGC at the optic nerve head, with subsequent death of the cell soma and axon within the retina, as well as Wallerian degeneration of the myelinated axon toward the brain. In human eyes, characteristic rearrangement of the connective tissues of the optic nerve head distinguish glaucomatous optic neuropathy from other disorders of the optic nerve.

2) "Elevated intraocular pressure (IOP) is the most common risk factor." Half or more of those with open angle glaucoma (the most common type) do not have "elevated" IOP, but the prevailing IOP in their eyes is associated with sufficient injury to RGC that damage happens. The risk factor is the level of the IOP, not elevated IOP.

3) "Current therapies target reduction of IOP, but irreversible RGC death continues even after IOP is normalized."

a)The aim of glaucoma therapy is not to "normalize" the IOP, but to lower it sufficiently from baseline (even when baseline is normal) to reduce progressive worsening to a low level.

b) The majority of IOP-lowered glaucoma eyes (OAG) have minimal further worsening as shown by large studies from clinical data (Chauhan et al). Some eyes require more lowering than others, as their worsening rate is substantially greater than average. A modest number of all OAG eyes have continued progressive RGC loss despite documented, maximal IOP lowering. The latter will be initial candidates from additional non-IOP lowering therapies.

4) "However, the difficulty of retaining microbeads at the angle of anterior chamber and of controlling the degree of aqueous outflow blockade results in a low success rate and high variabilities in the magnitude of IOP elevation and neurodegeneration." Published data show that 95% of microbead injected mouse eyes achieve significant IOP elevation (Cone et al). This is not a "low success rate" and is actually better than the 80% rate reported here for their oil injections. The variability in rates of RGC loss is more typical for human OAG eyes, and no model of glaucoma can expect to have the same damage rate in every rodent eye.

5) "We report that this treatment increases RGC somata and axon survival and significantly improves recovery of visual function". The word recovery might better be "protection of".

6) The mechanism of IOP elevation is almost surely not what is illustrated in the figure, nor is it similar to the human post-vitrectomy situation where oil blocks the pupil (in the absence of inferior iridotomy), leading in the human to secondary angle closure. Here, in the mouse, the mechanism is simply blockade of the meshwork and uveoscleral outflow directly by the oil. While this is not a fatal flaw for the model, it is silly to assume that the iris is going to bow forward with an anterior chamber full of oil (2 microliters of oil is more than the normal anterior chamber volume). If the authors wish to draw an analogy to human glaucoma due to silicone oil, the proper analogy is when emulsified oil blocks outflow after it moves into the anterior chamber.

7) The ASOCT images show the mouse iris separated in the "normal" eye by 40 degrees. In fact, in the mouse eye the iris is nearly touching the angle under normal conditions. These images could only be produced by abnormal pressure on the cornea during imaging.

8) The Figure 1 ASOCT images show the iris is 11 mm in length on the right side as pictured, then still 9 mm in length when "dilated". This does not match the larger pupil (but distorted shape) of the iris in the frontal view. The ASOCT images are far too small to properly interpret them.

9) The primary outcomes for the study should be: did the IOP rise (and how consistently) and after a reasonable time period did RGCs die – as well as showing that there was axonal transport blockade at the optic nerve head during the mid-portion of the period, demonstrating that the method kills RGCs as does human (and all other rodent models of ocular hypertension). These primary data are not only not convincing, the data are not even statistically tested. In Figure 1, we cannot tell how many eyes are included, nor whether the flags are standard deviations or standard errors. If as in the later figures these are standard errors, there is major variability in IOP (at least as presented after 15 minutes of anesthesia), not a consistent elevation as suggested by the authors. Furthermore, the bizarre factor of the model eyes having lower than control IOP for 10 minutes of anesthesia, then rising, is a huge error. We know from animal and human research that the earlier the IOP is measured after gas anesthesia, the closer the value is to the awake IOP. Waiting 15 minutes is not only completely impractical for lab personnel who need to measure large numbers of animals in a day, but why would we assume that the totally artificial, pupil-affected anesthetic state is what is representative of the awake mouse?

10) Consistency: how many animals were injected, how much oil was placed per eye, how much was still present at 1-3 days, how many animals didn't get an IOP elevation, were any animals not included in this report because they didn't have IOP increase?

11) "Pupillary size reached its maximum and IOP reached to its plateau about 12-15 minutes after induction of anesthesia with continuous isoflurane inhalation". So if we accept their theory that dilation allowed aqueous to reach the meshwork then IOP should go DOWN at that time point compare to initial anesthesia, not UP. No data on pupil size are presented to support this apparently backwards behavior compared to their pharmacological dilation.

12) Why did they inject saline into controls every week for 8 weeks? This would inevitably lower IOP by producing leaks of aqueous and confound the comparison.

13) There are many reports of a nearly identical injection into the anterior chamber using viscoelastic, mostly in rats, which is not cited or discussed. Oil would potentially be more long-lasting, and other similarities or differences should have been included.

14) Such massive RGC death is only seen in severe crush or transection injury in this time frame. While it is possibly true that this would give a bigger "signal" for neuroprotection studies, it also may make it harder for any protectant to block cell death with such severe and rapid damage. The authors should demonstrate that blood flow to the inner retina was not affected at early time points, as well as showing the retinal cross-sections that prove no inner 2/3 retinal damage (as might be expected from retinal artery compromise)-otherwise, this model is more an ischemia-reperfusion experiment.

15) No past AAV vector study (e.g. in rats by Martin) showed that control vector was "protective". It is more likely that the mice in this group either had less IOP elevation overall, or, the model has more variability than the authors believe.

16) No histology is shown that the effect mimics OAG by having axonal transport block at and just behind the ONH.

After the model is better presented, the other aspects of the research can be possibly considered. The authors might wish to focus on the model itself instead of adding in the vector studies.

[Editors’ note: what now follows is the decision letter after the authors submitted for further consideration.]

Thank you for submitting the revised version of your manuscript "Silicone oil-induced ocular hypertension in mouse models glaucomatous neurodegeneration and neuroprotection" together with your letter addressing the reviewers' comments. The manuscript has been re-reviewed and at this point we have a few additional queries, as detailed below.

1) If the mechanism of ocular hypertension in this model is via pupillary block, then how does one reconcile the fact that measured IOP is only elevated when the pupillary block is relieved? Despite the authors' explanation, it would be more convincing if the authors performed manometric readings in the anterior segment and posterior segment of the eye to prove this point.

2) The secondary analyses comparing later time points to week 1 (when OCT is still relatively unaffected); since the authors did not do a time 0 measurement of OKR and PERG. Please add into the manuscript the lack of progression of PERG compared to week 1, and how PERG amplitude reduction may be due to the presence of the SO itself or limitations of detection by PERG.

---

## [Author Response]

[Editors’ note: the author responses to the first round of peer review follow.]

We appreciate your effort in reviewing our manuscript. While we understand your concerns about the questions brought up by the reviewers, we think that the major concern can be addressed by a better explanation of the model itself. It reproduces the severe retinal neurodegeneration caused by ocular hypertension and is desperately needed for testing neuroprotection therapies in vivo and for studying neuronal responses to high IOP. Although it remains to be determined how this model resembles the more chronic and milder primary glaucoma, it faithfully mimics the acute secondary glaucoma in human patients. This is a new glaucoma model that differs greatly from previous models in many respects, including basic mechanism, method of induction, and outcomes. Initial questions and doubts are understandable, but the value of the model will become apparent once the data that it will generate are appreciated fully. We revised our manuscript to explain the model more clearly. We incorporate the reviewers’ comments and provide additional detailed information and new supporting data. Below we summarize the SO model in order to clarify the major questions as a whole. IOP rises or falls in response to two processes: aqueous inflow to the anterior chamber from the ciliary body in the posterior chamber and aqueous outflow through the TM at the angle of the anterior chamber. When aqueous inflow and outflow reach a steady state, normal IOP is maintained; when aqueous inflow exceeds outflow, IOP rises; when aqueous outflow exceeds inflow, IOP falls. Almost all previous glaucoma models increase IOP by decreasing aqueous outflow either by occluding the angle of the anterior chamber or by damaging the TM. In contrast, our model blocks aqueous inflow, which confines the aqueous to the posterior chamber and consequently increases the IOP of the posterior segment. The SO droplet touching the surface of the iris in combination with the large mouse lens forms a rigid barrier that seals the pupil and essentially separates the anterior chamber from the posterior segment.

This pupillary block has two results: 1) Aqueous produced by the ciliary body cannot flow into the anterior chamber and therefore accumulates and increases IOP in the posterior chamber; 2) The physical barrier formed by SO/iris/lens disconnects anterior chamber from posterior segment and, acts as a dam that keeps IOP low in the anterior segment while causing IOP to be high in the posterior segment where the aqueous accumulates. This explains why IOP always remains low in undilated SO eyes, even though the posterior IOP is greatly elevated (unfortunately posterior IOP cannot be measured directly). However, when the mouse pupil is dilated to the extent that it is no longer covered by the SO droplet (10-12 minutes after induction of anesthesia), the anterior and posterior chambers are re-connected, allowing the aqueous to flood into the anterior chamber quickly to increase the IOP in the anterior chamber. Initially, outflow from the TM is too slow to prevent the rise in IOP. However, since the TM continues to function normally, the high IOP increases aqueous outflow and eventually drives the IOP downward. Consistent with these mechanisms, we detected dynamic changes in IOP before and after pupil dilation, with corresponding low and high IOP. Furthermore, when we measured the IOP of a group of SO mice for an extended period of time (25-30 minutes after induction of anesthesia and 10-15 minutes after full pupillary dilation), the IOP decreased (new supplementary figure). We now include this new data to demonstrate two points: 1. SO increases IOP in the posterior segment by blocking the pupil and preventing aqueous inflow into the anterior chamber, but the increased IOP can only be detected after pupillary block is removed and aqueous inflow restored because the Tonolab can only measure IOP of the anterior segment. 2. The IOP eventually decreased after prolonged pupillary dilation, which indicates that TM continues to function normally and allow aqueous to pass through it. We have never observed small emulsified SO droplets in the mouse eyes and if invisible emulsified SO occluded TM, the IOP level should be high and unresponsive to pupil dilation, which is completely the opposite of our observations in this model. We think that this manuscript has been substantially improved and hope that it can be reviewed again.

Reviewer #1:[…] This model differs from other established models of experimental glaucoma, in that it has a high success rate of IOP elevation that can be easily predicted by SO diameter, severe cell loss, and minimal tissue damage and inflammation. Due to its ease of induction, rapid and severe development of glaucomatous neurodegeneration, this model will be valuable in investigating molecular mechanisms and screening drug compounds.- What implications does mechanisms of secondary pupillary block angle-closure glaucoma have on primary open angle glaucoma? It would be worth mentioning that this type of postoperative secondary glaucoma is rare since surgeons prophylactically create peripheral iridotomies. Indeed, secondary glaucoma caused by SO is more frequently the result of emulsification of SO leading to very small bubbles lodged in the angle. Discussion, fourth paragraph, the last sentence is a bit overstated, and a clearer discussion of how this model differs from human glaucoma and POAG would be appreciated.

We appreciate reviewer 1’s recognition that our model “differs from other established models” and “valuable in investigating molecular mechanisms and screening drug compounds”. Here we primarily replicate secondary glaucoma caused by pupillary block and provide an acute glaucoma model which is valuable for studying the effect of sustained high IOP on RGCs and optic nerve. We agree with the reviewer that preventive iridotomy significantly lowers down the risk of postoperative secondary glaucoma caused by SO, and we have added this comment to the Discussion. We have also modified the sentence and added comments on the differences between our model and human POAG, which is more chronic and milder.

- How does the effect of IOP elevation in the posterior chamber differ from the anterior chamber? We had difficulty understanding the naming of the "SOHU" model. Is there known undermeasurement of IOP when the anterior chamber is filled with silicone oil?

We suspect that in this model the large lens of the mouse eye combines with the iris, ciliary body and SO droplet to form a rigid barrier that essentially separates the anterior chamber from the posterior segment. This disconnection allows different IOP levels to exist in the anterior and posterior chambers. Because the volume of aqueous inflow is small, IOP in the anterior chamber remains low until pupillary dilation reconnects the anterior and posterior chambers and aqueous flooding into the anterior chamber produces a detectable elevation of IOP.

- A more detailed explanation in the Materials and methods for IOP measurement would be appreciated. Was pupillary block removed before measuring IOP (i.e. eyes were dilated?)? Was IOP measured at the same time of day? The authors are trying to establish this model as a model of increased IOP that leads to neurodegenerative changes. IOP measurements fluctuate daily and reporting IOP measurements only once a week seems insufficient. Do light/dark cycles affect IOP elevation since pupillary block is relieved upon dilation that would occur in dark?

We have added a more detailed description of IOP measurement to Materials and methods (“IOP measurement”). Mice are anesthetized with a sustained flow of isoflurane (3% isoflurane at 2 L/minute mixed with oxygen) when we take measurement with the TonoLab tonometer every 5 minutes for the time periods indicated in Figure 1C. The pupil enlarges slowly and is fully dilated at 10-12 minutes after induction of anesthesia; IOP in the SO eyes correspondingly increases as the pupil enlarges and plateaus when the pupil is fully dilated. In another group of mice, we extended the IOP measurements until 30 minutes after induction. IOP decreases 25-30 minutes after the onset of anesthesia, which is 10-15 minutes after the pupil become fully dilated, presumably because the TM functions normally and clears the aqueous efficiently from the anterior chamber.

We can only detect the actual IOP after the pupillary block is removed, the pupil size is larger than the SO droplet, and the aqueous floods into the anterior chamber from the posterior chamber. We always measure IOP in the morning. We did not measure IOP at night, but we doubt that dark adaptation would enlarge the pupil enough to overcome the SO blocking effect.

We measure IOP once a week because each measurement requires dilating the pupil and releasing the accumulated aqueous from the posterior of the eye, which reduces IOP. To minimize IOP fluctuation and maintain a stable IOP elevation, we try to minimize the measurement.

- Are OKR and PERG reliable functional assessments if visual acuity is impaired due to difference in refractive index of SO? Is the baseline "0" (where data is normalized to 100%) a measurement made at baseline in which neither eye has been perturbed, or is it measured right after SO and saline injections? The latter would be the correct control.

We agree with the reviewer that the difference in refractive index between SO and aqueous may induce an artifactual effect on the readouts of OKR and PERG, since both depend on light stimulation. We commented on this issue in the original Discussion: “One caveat, however, is that SO itself in anterior chamber may blur vision or affect the visual function assays because its optical characteristics differ from those of aqueous humor. These differences may cause early decreases in visual acuity and PERG amplitude at 1wpi, when OCT imaging, which does not depend on the transparency of anterior segment of the eye, shows no significant morphological degeneration.”

We measure baseline before any injection and always normalize to the contralateral eyes to minimize variation. We agree with the reviewer that it would be better to acquire the baseline immediately after injection, but in practice this is difficult because the condition of the cornea is not optimal for OKR or PERG measurement immediately after anterior chamber injection, and cornea recovery from the injection injury takes time.

- The manner in which the data is presented for OCT, OKR, and PERG in Figure 2 do not show that there is progression over time. This data needs to be presented more clearly, since in the text the authors argue that there is progressive thinning (OCT), progressive loss of vision (OKR), and progressive decrease in the P1-N2 amplitude (PERG) with time. If there was no progression for OKR and PERG, then it suggests that the SO bubble itself is the cause for the decreased amplitudes.

We agree with the reviewer that the sharp drop of OKR and PERG at 1wpi may be due to SO’s optical effect, rather than a real change in visual function. There is also a small but significant decrease of OCT and OKR in the later weeks, however, and we have added a comparison of the results at 1 wpi *vs* those at 5 and 8 wpi to show this. The PERG demonstrates a trend toward a further decrease between 1 wpi and later times, but this decrease does not reach statistical significance, perhaps because of the limitations in detection of PERG.

- Figure 3, middle panel. What portion of the flat mount retina is represented (central/peripheral)? The authors should comment on why the peripheral retina appears to have much more significant loss than centrally. Also, how were the 6-9 fields "randomly" selected to quantify RBPMS? It seems that whether one chose fields from the center vs. periphery would play a large role in the quantification.

We only quantify peripheral RGCs because RGCs in the central retina are too densely packed to be reliably counted. The 6-9 fields imaged from peripheral retina are to cover the bulk of this part of the retina. The middle panel of Figure 3 presents representative images from peripheral retinas. Because the density of peripheral RGCs is much lower than central, the loss of RGCs is more obvious in peripheral than central retina, at least by visual inspection. However, until we can reliably count RGCs in the central retina, we cannot definitively determine whether central or peripheral RGCs die first or faster.

Reviewer #2:This manuscript reports the development of a new method of inducing IOP elevation in mice by intracameral injection of silicone oil (SO). It has provided data to show that SO injection caused reduced visual acuity and GCC thickness, deficits in PERG, and the loss of retinal ganglion cells and axons. Modifying ER stress pathways by AAV delivery of CHOP shRNA and XBP-1s significantly prevented RGC and axon loss. It also showed that flushing out the SO by anterior chamber injection of saline allows the IOP level returning to normal. The model of SO-induced ocular hypertension is novel and can be very interesting. However, the manuscript presents several issues that need to be addressed:1) TanoLab assessment for IOP in this model may not reflect the real IOP value in the back of the eye. As it is stated in the manuscript: "Because of the unique feature of pupillary block associated with SOHU, the IOP is elevated in the posterior part of the eye, but not in the anterior chamber". TanoLab measures the pressure of the anterior chamber, thus is incapable of assessing the actual IOP value in the posterior part of the eye, which is likely to be far higher than it was detected by TanoLab in this case.

We agree with the reviewer’s comments that the IOP of the posterior part of the eye may be much higher than that detected by the Tonolab tonometer after pupil dilation. Indeed we have speculated that this elevation in IOP is the explanation for why degeneration occurs more rapidly and is more severe with SO injection than in other models.

2) The rate of RGC loss in this model is excessive. As data showed in Figure 3, elevation of IOP led to nearly 90% RGC loss (12% RGC survival) in 8 weeks; this represents a very severe damage, similar to the situation of complete transection of the optic nerve, and it does not replicate the most cases of glaucoma.

We agree with the reviewer that this method produces a much faster and more severe glaucomatous neurodegeneration than other models. This model replicates acute secondary glaucoma, which is notably rapid and severe. Because it generates neuronal responses to maintained high IOP, the method can be used experimentally to test neuroprotective therapies in a short period of time.

3) The manuscript showed either a large neuroprotective effect by control AAV injection (as suggested by the authors) or inconsistent neural damage from time to time in the SOHU model. In Figure 3, injection of SO ended up with 12% RGC survival by 8 weeks, while in Figure 4, counts of survival RGCs were ~60% by 8 weeks. Such a huge neuroprotective effect by control AAV injection has not been observed or reported by others. If the authors claimed that they found it in other optic neuropathy models, the data need to be provided.

We repeated the experiments twice and obtained a similar AAV effect on RGC survival in this model. We have observed similar outcomes in the optic nerve crush and EAE models. We suspect that AAV injection itself may prime RGCs to be more resistant to other damage. The effect of AAV is not the major point of this paper, however, and as reviewer 3 suggested, we have removed it from this revised manuscript to keep the focus on the novel model.

Reviewer #3:The proposal that silicone oil injection may be a useful mouse high pressure model is reasonable, and if it is true that the cornea stays clear, this would be an advantage over other mouse models. However, every model in mice shows corneal enlargement (and axial length increase) with IOP in the range reported here, so further studies of clarity need to be presented. Removal of the oil to lower IOP is potentially useful adjunct, if one wishes to study various periods of IOP elevation, then allow time for further events at normal IOP. Unfortunately, the work as presented does not adequately show the true primary outcomes definitively and objectively.

We appreciate reviewer 3’s recognition that our model “may be useful” and “potentially useful”. We have thoroughly revised the manuscript to better present our results and our discussion of these results.

1) "Glaucoma is a neurodegenerative disease characterized by optic neuropathy with thinning of the retinal nerve fiber layer (RNFL) followed by progressive retinal ganglion cell (RGC) degeneration (Quigley, 1993; Quigley et al.,1995; Livvy et al., 2005; Howell et al., 2007; Weinreb and Khaw, 2004; Calkins, 2012; Burgoyne, 2011; Nickells et al., 2012; Jonas et al., 2017)." This is an incorrect and inexact definition of glaucomatous optic neuropathy. Glaucoma is characterized by injury to the axons of RGC at the optic nerve head, with subsequent death of the cell soma and axon within the retina, as well as Wallerian degeneration of the myelinated axon toward the brain. In human eyes, characteristic rearrangement of the connective tissues of the optic nerve head distinguish glaucomatous optic neuropathy from other disorders of the optic nerve.2) "Elevated intraocular pressure (IOP) is the most common risk factor." Half or more of those with open angle glaucoma (the most common type) do not have "elevated" IOP, but the prevailing IOP in their eyes is associated with sufficient injury to RGC that damage happens. The risk factor is the level of the IOP, not elevated IOP.3) "Current therapies target reduction of IOP, but irreversible RGC death continues even after IOP is normalized."a)The aim of glaucoma therapy is not to "normalize" the IOP, but to lower it sufficiently from baseline (even when baseline is normal) to reduce progressive worsening to a low level.b) The majority of IOP-lowered glaucoma eyes (OAG) have minimal further worsening as shown by large studies from clinical data (Chauhan et al). Some eyes require more lowering than others, as their worsening rate is substantially greater than average. A modest number of all OAG eyes have continued progressive RGC loss despite documented, maximal IOP lowering. The latter will be initial candidates from additional non-IOP lowering therapies.4) "However, the difficulty of retaining microbeads at the angle of anterior chamber and of controlling the degree of aqueous outflow blockade results in a low success rate and high variabilities in the magnitude of IOP elevation and neurodegeneration." Published data show that 95% of microbead injected mouse eyes achieve significant IOP elevation (Cone et al). This is not a "low success rate" and is actually better than the 80% rate reported here for their oil injections. The variability in rates of RGC loss is more typical for human OAG eyes, and no model of glaucoma can expect to have the same damage rate in every rodent eye.5) "We report that this treatment increases RGC somata and axon survival and significantly improves recovery of visual function". The word recovery might better be "protection of".

Response to reviewer 3’s comments 1-5: We appreciate the reviewer’s instructive comments on glaucoma and we have modified the Introduction accordingly.

6) The mechanism of IOP elevation is almost surely not what is illustrated in the figure, nor is it similar to the human post-vitrectomy situation where oil blocks the pupil (in the absence of inferior iridotomy), leading in the human to secondary angle closure. Here, in the mouse, the mechanism is simply blockade of the meshwork and uveoscleral outflow directly by the oil. While this is not a fatal flaw for the model, it is silly to assume that the iris is going to bow forward with an anterior chamber full of oil (2 microliters of oil is more than the normal anterior chamber volume). If the authors wish to draw an analogy to human glaucoma due to silicone oil, the proper analogy is when emulsified oil blocks outflow after it moves into the anterior chamber.

Our results lead us to respectfully disagree with these comments. The SO droplet clearly blocks the pupil, as demonstrated in the images and videos, and we have never seen small emulsified SO droplets in any of the mouse eyes. Also, if the mechanism of IOP elevation is TM blockade, then the IOP level should be high even before pupillary dilation and not increase after dilation, which is totally opposite to our observation. Moreover, all of our other results indicate that TM functions normally in SO eyes, including: pupillary dilation first increases anterior chamber IOP and then reduces it; SO removal reduces IOP to normal; and a small SO droplet does not cause sustained IOP elevation.

Additionally, the available literature conflicts with the suggestions that the volume of the normal mouse anterior chamber is ≤ 2µl. According to J Ocul Pharmacol Ther. 2016 Jan 1; 32(1): 28–37. “Species Differences in the Geometry of the Anterior Segment Differentially Affect Anterior Chamber Cell Scoring Systems in Laboratory Animals”, anterior chamber volume is 7 µl. We injected 2 µl SO.

7) The ASOCT images show the mouse iris separated in the "normal" eye by 40 degrees. In fact, in the mouse eye the iris is nearly touching the angle under normal conditions. These images could only be produced by abnormal pressure on the cornea during imaging.8) The figure 1 ASOCT images show the iris is 11 mm in length on the right side as pictured, then still 9 mm in length when "dilated". This does not match the larger pupil (but distorted shape) of the iris in the frontal view. The ASOCT images are far too small to properly interpret them.

We suspect that reviewer 3 has mistaken the edge of the lens for a part of the iris. The pupil is wide open after dilation. Our high-definition images can be enlarged to view anatomical details.

9) The primary outcomes for the study should be: did the IOP rise (and how consistently) and after a reasonable time period did RGCs die – as well as showing that there was axonal transport blockade at the optic nerve head during the mid-portion of the period, demonstrating that the method kills RGCs as does human (and all other rodent models of ocular hypertension).

We agree with the reviewer and follow exactly the suggested way of presenting our results: IOP elevation that causes RGC degeneration. Concerning the optic nerve head (ONH): because mice lack a lamina cribrosa, the likelihood of finding that high IOP causes ONH compression is small, if not impossible, as it is for finding optic nerve disc “cupping”, and no rodent model of glaucoma has convincingly demonstrated ONH compression before. I will be interesting to test these ideas in non-human primate, which possesses a lamina cribrosa structure similar to that in human.

These primary data are not only not convincing, the data are not even statistically tested. In figure 1, we cannot tell how many eyes are included, nor whether the flags are standard deviations or standard errors. If as in the later figures these are standard errors, there is major variability in IOP (at least as presented after 15 minutes of anesthesia), not a consistent elevation as suggested by the authors. Furthermore, the bizarre factor of the model eyes having lower than control IOP for 10 minutes of anesthesia, then rising, is a huge error. We know from animal and human research that the earlier the IOP is measured after gas anesthesia, the closer the value is to the awake IOP. Waiting 15 minutes is not only completely impractical for lab personnel who need to measure large numbers of animals in a day, but why would we assume that the totally artificial, pupil-affected anesthetic state is what is representative of the awake mouse?

Figure 1 for SO > 1.5mm, n=17; SO ≤ 1.5mm, n=6. All data are represented as mean ± SEM and since IOP is obviously higher in the SO eyes, we did not use a star to indicate statistical significance. We use “consistent” to refer to IOP elevation compared to the contralateral eye, not to mean that all eyes have the same IOPs.

The manuscript contains a detailed presentation of the techniques used to measure IOP in this model and a full consideration of the issues related to pupillary block-induced ocular hypertension. We have also dealt with these techniques and issues in previous answers to reviewer 1’s questions. As reviewer 3 noted, we devoted a great deal of effort and considerable thought to IOP measurement, and we hope that he/she agree that, although the procedure is cumbersome and requires two dedicated staff members, it is the only way to obtain the full picture and reliable IOP data. Importantly, one valuable feature of this model is that, because the IOP elevation is very stable and predictable from the size of the SO droplet, after the verification stage, frequent IOP measurement will not be necessary.

10) Consistency: how many animals were injected, how much oil was placed per eye, how much was still present at 1-3 days, how many animals didn't get an IOP elevation, were any animals not included in this report because they didn't have IOP increase?

We include animal numbers in every figure legend. Each eye received about 2 µl SO. Most of the eyes (80%) contained a SO droplet size > 1.5mm and about 20% contained smaller SO droplets, but the size of all the droplets was largely maintained for at least as long as the periods that we examined. We used 1.5mm SO droplet in later experiments to exclude animals that would not have a reliable IOP elevation at 1 week after SO injection. We include every animal with an SO droplet size > 1.5mm, because all of these have an IOP elevation that is maintained for at least as long as we examined them.

11) "Pupillary size reached its maximum and IOP reached to its plateau about 12-15 minutes after induction of anesthesia with continuous isoflurane inhalation". So if we accept their theory that dilation allowed aqueous to reach the meshwork then IOP should go DOWN at that time point compare to initial anesthesia, not UP. No data on pupil size are presented to support this apparently backwards behavior compared to their pharmacological dilation.

As we explained in response to previous comments, IOP increase or decrease depends on two processes: aqueous input from the posterior segment and outflow/clearance through the TM. When aqueous fluid inflow and outflow are equal, a steady-state IOP is maintained; when inflow exceeds outflow, IOP rises; when outflow exceeds inflow, IOP decreases. We can capture the dynamic changes of IOP in SO mouse eyes because, with the animals under anesthesia, we can measure IOP multiple times over an extended period of time. However, similar measurements cannot be easily obtained in human patients. It is plausible that, even in human, the IOP will increase immediately after aqueous floods into the anterior chamber if the clearance rate cannot match the speed of influx. We suspect that the same dynamic changes of IOP that we observed in mice will also occur in patients with pupillary block, but it is difficult to capture these changes in their entirety if clearance is more rapid through the human TM or because it is not feasible to monitor IOP continually for an extended period of time in humans.

12) Why did they inject saline into controls every week for 8 weeks? This would inevitably lower IOP by producing leaks of aqueous and confound the comparison.

We measure IOP weekly, we inject saline as a control only once. We modified the sentence to make it clear.

13) There are many reports of a nearly identical injection into the anterior chamber using viscoelastic, mostly in rats, which is not cited or discussed. Oil would potentially be more long-lasting, and other similarities or differences should have been included.

As the reviewer suggests, we now discuss the difference between our pupillary block model and other models that use repeated viscoelastic injections for TM occlusion.

14) Such massive RGC death is only seen in severe crush or transection injury in this time frame. While it is possibly true that this would give a bigger "signal" for neuroprotection studies, it also may make it harder for any protectant to block cell death with such severe and rapid damage. The authors should demonstrate that blood flow to the inner retina was not affected at early time points, as well as showing the retinal cross-sections that prove no inner 2/3 retinal damage (as might be expected from retinal artery compromise)-otherwise, this model is more an ischemia-reperfusion experiment.

We agree with the reviewer’s point that high IOP may decrease blood flow and thus indirectly injury the retina. As in the clinic, acute IOP elevation may cause ischemia. Moreover, multiple studies have reported significantly decreased blood flow in ophthalmic artery and choroidal circulation in patients with POAG and we cited those references in the Discussion. By re-evaluation of our OCT images revealed that, in dramatic contrast to significant thinning of the GCC, there is no significant thinning of the outer nucleus layer, indicating no significant photoreceptor death (Author response image 1). We are also doing another batch of SOHU model and will determine histologically the thickness of different layers of retina with cross-sections to demonstrate potential damages in other layers of retina.

15) No past AAV vector study (e.g. in rats by Martin) showed that control vector was "protective". It is more likely that the mice in this group either had less IOP elevation overall, or, the model has more variability than the authors believe.

We appreciate the reviewer’s suggestion and we have removed this figure to focus on the model itself.

16) No histology is shown that the effect mimics OAG by having axonal transport block at and just behind the ONH.

Please see our response to reviewer #3, comment #9.

After the model is better presented, the other aspects of the research can be possibly considered. The authors might wish to focus on the model itself instead of adding in the vector studies.

[Editors’ note: the author responses to the re-review follow.]

Thank you for submitting the revised version of your manuscript "Silicone oil-induced ocular hypertension in mouse models glaucomatous neurodegeneration and neuroprotection" together with your letter addressing the reviewers' comments. The manuscript has been re-reviewed and at this point we have a few additional queries, as detailed below.1) If the mechanism of ocular hypertension in this model is via pupillary block, then how does one reconcile the fact that measured IOP is only elevated when the pupillary block is relieved? Despite the authors' explanation, it would be more convincing if the authors performed manometric readings in the anterior segment and posterior segment of the eye to prove this point.

We agree with the reviewer that it would be best to be able to detect the intraocular pressures in the anterior chamber and posterior segment separately. We have tested it at the very beginning of developing this model, unfortunately, we can never get reliable manometric pressure reading by poking the detection needle into the vitreous chamber of the mouse eye. We suspect the viscous vitreous blocks the needle to prevent any pressure detection. We are not aware anyone has been successfully measured the pressure of the posterior segment of the mouse eye. We indeed can detect relative higher IOP in the anterior chamber of SO eye compared to naïve eye (Author response image 2), although again, we are not very confident about the manometric reading and decide not including this data in the manuscript.

**Author response image 2. respfig2:** 

2) The secondary analyses comparing later time points to week 1 (when OCT is still relatively unaffected); since the authors did not do a time 0 measurement of OKR and PERG. Please add into the manuscript the lack of progression of PERG compared to week 1, and how PERG amplitude reduction may be due to the presence of the SO itself or limitations of detection by PERG.

We agree with the reviewer and we added the following sentence in the “Results” section: “However, that the lack of progression of PERG amplitude reduction suggests the SO itself may affect the light stimulation and PERG signal or the limitations of detection by PERG.”